# Deep Augmentation: Self-Supervised Learning with Transformations in Activation Space

## Abstract

We introduce Deep Augmentation, an approach to implicit data augmentation using dropout or PCA to transform a targeted layer within a neural network to improve performance and generalization. We demonstrate Deep Augmentation through extensive experiments on contrastive learning tasks in NLP, computer vision, and graph learning. We observe substantial performance gains with Transformers, ResNets, and Graph Neural Networks as the underlying models in contrastive learning, but observe inverse effects on the corresponding supervised problems. Our analysis suggests that Deep Augmentation alleviates co-adaptation between layers, a problem exhibited by self-supervised learning where ground truth labels are not available. We use this observation to formulate a method for selecting which layer to target; in particular, our experimentation reveals that targeting deeper layers with Deep Augmentation outperforms augmenting the input data. The simple network- and modality-agnostic nature of this approach enables its integration into various machine learning pipelines.

## 1 Introduction

Self-supervised learning is a paradigm shift in machine learning that creates representations and pre-trained models without relying on human-annotated labels. It has revolutionized domains including computer vision (Chen et al., 2020), natural language processing (Devlin et al., 2019), learning on graphs (Zhu et al., 2021), speech processing (Oord et al., 2016), and genomics (Zaheer et al., 2020). Contrastive learning (Oord et al., 2018; Chen et al., 2020), a popular strategy within self-supervised learning, demonstrates exceptional results. This approach leverages data augmentations to create complementary pairs of samples that preserve semantic structure (Shorten & Khoshgoftaar, 2019) and artificially expand the training data.

Currently, effective design of data augmentations necessitates a deep understanding of the domain or dataset. In image processing, data augmentations are built from image processing operations like cropping and blurring (Chen et al., 2020), while in NLP, techniques like masking and synonym replacement dominate (Gao et al., 2021).

In this work, we introduce Deep Augmentation (Figure 1), a network- and modality-agnostic method for augmentation in higher layers of neural networks (NNs) using dropout (Srivastava et al., 2014), principal component analysis (PCA) (F.R.S., 1901), and an optional stop-gradient operation. These transformations in high-dimensional activation spaces improve performance and generalization in computer vision with ResNets (He et al., 2016), NLP with Transformers (Vaswani et al., 2017), and graphs with Graph Neural Networks (Kipf & Welling, 2017). Unlike other methods, Deep Augmentation does

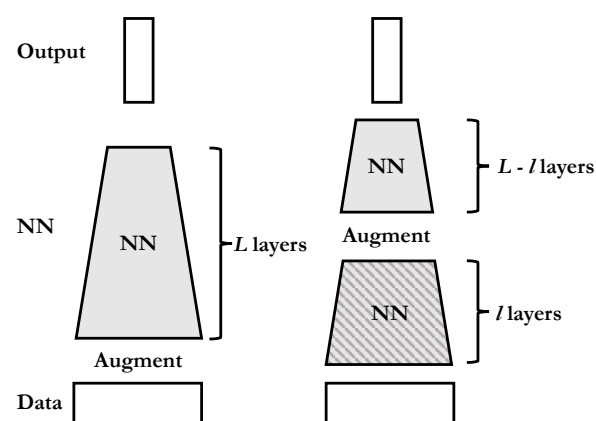

Figure 1: Left: Traditional augmentation. Right: Deep Augmentation at layer $l$.

not require expert-designed and handcrafted augmentations and does not rely on supervised labels, making it versatile and broadly applicable.

Our main contributions are as follows:

- We demonstrate that layer-targeted dropout, with and without a stop-gradient operation, enhances contrastive learning performance. It enhances sentence embeddings without data augmentations and images/graph embeddings with data augmentations, with pronounced effect when applied to higher layers.
- Conversely, we observe an inverse effect in supervised learning contexts, potentially due to reduced co-adaptation between layers, akin to an information bottleneck, already naturally occurring when directly training on downstream tasks.
- As an additional strategy for modality-agnostic augmentation, we show that a simple PCA-inspired augmentation can be substituted for dropout with similar performance.
- Based on the observation that our techniques can ameliorate co-adaptation among network layers and their resultant latent features, we propose a procedure for selecting the target layer for Deep Augmentation.
- We substantiate the claims above through extensive ablation studies.

## 2 Related Work

**Self-Supervised Learning.** Self-supervised learning (Chen et al., 2020; Grill et al., 2020; Caron et al., 2020; Chen & He, 2021; Rani et al., 2023) leverages unlabeled data to train representations transferable to downstream tasks. This approach involves a pre-training task using pseudo-labels for training the model. Given the abundance of unlabeled over labeled data, this method allows for training larger models on extensive datasets with reduced overfitting, making it popular for high-quality, transferable representations.

**Data Augmentation.** Constrained by the scarcity of labeled data, supervised learning benefits from augmentations that preserve semantic integrity while enhancing data volume, performance, and generalization (Lecun et al., 1998). While supervised and contrastive learning often employ similar augmentations (Chen et al., 2020), supervised learning uniquely uses label-based techniques like Mixup (Zhang et al., 2018) to generate synthetic examples. In self-supervised learning, augmentation typically occurs in the input space, the initial layer in Deep Augmentation. Despite its interpretability, this layer may not always yield the most compelling features for augmentation.

**Higher-Layer Features.** Zhou et al. (2022); Hinton (2023) highlight the significance of capturing semantic granularity in representations to enhance their applicability to downstream tasks. Given that neural networks inherently learn hierarchical features (Bilal et al., 2018; Brüel Gabrielsson & Carlsson, 2019), augmenting intermediate activations fosters invariance across varying data perspectives, ranging from input space to abstract interrelations in deeper layers.

**Higher-Layer Augmentation.** Prior work investigates latent space augmentations via interpolation methods like Manifold Mixup (Verma et al., 2018), applying Mixup to hidden layer outputs, and linear interpolation for image classification (DeVries & Taylor, 2017). MODALS (Cheung & Yeung, 2021) integrates these using reinforcement learning.

**Dropout as Augmentation.** *Dropout* is a key strategy in deep learning (Labach et al., 2019; Salehin & Kang, 2023) that can be interpreted as generic data augmentation, often used in supervised learning Bouthillier et al. (2015). Recent work on sentence embedding suggests its possible application in contrastive learning (Gao et al., 2021). Its widespread use suggests the necessity of examining of dropout's role in augmenting intermediate activations. This includes assessing its distinct utility and exploring potential variations in effectiveness between supervised and contrastive learning. Previous work typically views dropout as data augmentation when applied across *all* layers, but unjustifiably assumes that all layers are similarly amenable to dropout augmentation. We show that targeting certain layers is key for successful dropout augmentation in contrastive learning.

**Stop-Gradient & Information Collapse.** Chen & He (2021) implement stop-gradient in Siamese networks, demonstrating its capability to facilitate unsupervised visual representation learning without negative sample pairs, large batches, or momentum encoders. Their findings reveal that, although collapsing solutions (Jing et al., 2022) exist, stop-gradient plays a crucial role in averting such outcomes. Their comparisons include one experiment applying stop-gradient to one side of the positive pairs in SimCLR (Chen et al., 2020), slightly decreasing performance. Our work extends this inquiry, conducting broader experiments across various domains, focusing on specific network layers,

integrating augmentations, and applying these methods to supervised learning scenarios. In contrast to their findings, we observe performance improvements in our experiments.

There are intriguing connections between information theory and the desirability of certain information collapse in self-supervised learning and supervised learning (Tian et al., 2020; Shwartz-Ziv & LeCun, 2023), as exemplified by information bottlenecks (Tishby & Zaslavsky, 2015). We contribute to this discussion by providing evidence that Deep Augmentation impacts forms of collapse, such as co-adaptation between layers and uniformity in latent features, potentially explaining the divergent effects of Deep Augmentation on contrastive and supervised learning.

**Analysis of Representation Learning.** Wang & Isola (2020) identify two key quality measures for representation achieved by contrastive learning: (1) alignment of features from positive pairs, and (2) uniformity of the induced distribution of (normalized) features on the hypersphere. Kornblith et al. (2019) presents a similarity index that measures the relationship between representational similarity matrices. It is equivalent to centered kernel alignment (CKA) and reliably identifies correspondences between representations in NNs trained from different initializations.

## 3 Method

### 3.1 Preliminaries

Contrastive learning aims to learn representations by attracting semantically similar pairs while distancing non-pairs. For a dataset $X = \{x_1, \ldots, x_N\}$, it forms pairs $\mathcal{D} = \{(x_i^1, x_i^2)\}_{i=1}^m$, where $x_i^1$ and $x_i^2$ are different but semantically similar to $x_i \in X$. Pair construction in contrastive learning is pivotal as it defines the learned invariances. It typically involves generating dual views of a sample through random transformations, like cropping, flipping, distortion, and rotation in images.

More specifically, a random augmentation is independently applied to each sample. In this sense, let $Z \sim \mu$ be a random variable where $\mu \in \text{Prob}(\Omega)$ for some space $\Omega$. $\Omega$ can be discrete, e.g. cropping size, or continuous, e.g. blurring variance. Let $A : \mathbb{R}^d \times \Omega \to \mathbb{R}^d$ be an augmentation function, $B \subset X$ a randomly drawn batch, and $z_i^1, z_i^2 \sim \mu$ be a pair of samples. The features of the augmented pairs are defined as $h_i^j := f_\theta(A(x_i, z_i^j))$ for $j \in \{1, 2\}$, where $f_\theta$ is a NN with learnable parameters $\theta$.

The InfoNCE (Chen et al., 2020) loss for $B$ is then:

$$l(\theta; B) = \frac{1}{|B|} \sum_{i=1}^{|B|} \log \frac{e^{\text{cosine-sim}(h_i^1, h_i^2)/\tau}}{\sum_{j=1}^{|B|} e^{\text{cosine-sim}(h_i^1, h_j^2)/\tau}}. \tag{1}$$

This loss encourages $f_\theta$ to be invariant to $A$ and $\{\frac{h_i}{||h_i||} \mid x_i \in X\}$ to be uniformly distributed (Wang & Isola, 2020).

### 3.2 Deep Augmentation

A NN $f_\theta$ processes data by composition along its layers. Assuming $f_\theta$ has $L$ layers, we let $f_\theta^{a,b}$, $-1 \leq a \leq b < L$ be the operations from layer $a$ to $b$, where $a = -1$ represents the data input and $a = b$ represents the identity operation. Then, for any $-1 \leq l < L$, we can decompose $f_\theta = f_\theta^{l+1,L-1} \circ f_\theta^{-1,l}$. For example, only augmenting the input data can be notated $f_\theta(A(x_i, z_i^j)) = f_\theta^{0,L-1} \circ A(f_\theta^{-1,-1}(x_i), z_i^j)$.

In this work, we investigate the setup

$$g_\theta^l := f_\theta^{l+1,L-1} \circ A(f_\theta^{-1,l}(x_i), z_i^j) \tag{2}$$

for $-1 \leq l < L$; see Figure 1 for a diagram. Immediately, we recognize that our work is simplified if $A$ has certain properties: (1) layer-agnostic (we can use any $l$ without changing $A$), (2) network-agnostic (we can change the architecture of $f_\theta$ without changing $A$), and (3) modality-agnostic (we can change the input data without changing $A$). We achieve these properties by choosing a simple dropout operation for $A$, but also test, in Section 5, an alternative PCA-inspired augmentation.

We study which values of $l$ yield the best representation $g_\theta^l$ as judged by performance on downstream tasks. In settings where we use Deep Augmentation together with input data augmentations, Deep Augmentation is applied in composition with the input-data augmentation.

**PCA Augmentation.** We evaluate whether dropout is the sole effective augmentation technique in Deep Augmentation. In contrast to dropping individual neurons as in dropout, we experiment with subtracting a principal component in the feature space. More formally, for a mini-batch with indices denoted as $I_b = 1, 2, \ldots, K$, the augmentation for a sample $x_i, i \in I_b$ is defined as follows:

$$A_p(x_i, z_i^j) = x_i - PC(\{x_k : k \in I_b\})[:, p]$$

where $PC$ projects onto the principal components of the set and arranges them in descending order of their eigenvalues. The batch is centered prior to this operation and re-centered afterwards. This method is hereafter referred to as PCA.

**Stop-Gradient.** In the context of Equation 2, when $l > -1$, we move beyond the standard setting of $l = -1$, introducing learnable layers *prior* to augmentation $A$. Here, we can apply the stop-gradient operation, halting gradient propagation below the targeted layer during augmentation (Chen & He, 2021). This allows us to examine the impact of learning invariance to forthcoming augmentations versus invariance to those already applied.

**Sample 50% of Batch.** Deep Augmentation is applied by default to a random 50% of samples in each batch. This enhances the variability of augmentations and correspondence between training and evaluation, by ensuring that a subset of samples in each batch remains unaugmented. Additionally, this facilitates the training of all neural network layers while applying stop-gradient, as it will not be applied to half the batch. See Appendix A.4 for discussion and ablation study.

## 4 Main Results

We demonstrate Deep Augmentation's potential to significantly boost contrastive learning effectiveness across vision (Table 3), natural language processing (Table 2), and graph-based learning (Table 4). These enhancements stem from the optimal configuration of targeted dropout and stop-gradient, not necessarily representing peak performance under all possible Deep Augmentation hyperparameters.

In sentence embedding, we adopt the methodology of Gao et al. (2021), pre-training a BERT transformer (Devlin et al., 2019) on $10^6$ randomly sampled sentences from the English Wikipedia. Hyperparameters are compared on the STS-B development set (Cer et al., 2017) and evaluations are conducted on seven standard semantic textual similarity (STS) tasks (Agirre et al., 2012; Cer et al., 2017; Marelli et al., 2014). For vision, a ResNet model (He et al., 2016) is used on natural image datasets including CIFAR10, CIFAR100, and ImageNet100, a subsampled version of ImageNet (Deng et al., 2009). In graph contrastive learning, we follow Zhu et al. (2021) in using a graph convolutional network (Kipf & Welling, 2017). Hyperparameter selection for each dataset—COLLAB and IMBD-Multi (Yanardag & Vishwanathan, 2015), NCI1 (Wale & Karypis, 2006), and PROTEINS (Borgwardt et al., 2005)—uses a validation set, with evaluations on a separate test set.

Each benchmark employs its respective standard contrastive learning framework (SimCSE (Gao et al., 2021) for sentence embeddings, SimCLR (Chen et al., 2020) for images, and GCL (Zhu et al., 2021) for graphs), incorporating dropout across all layers with fine-tuned dropout rates. This highlights the added value of Deep Augmentation in enhancing learning across datasets.

Table 1: Contrastive Learning on Sentence Embeddings with Transformer. Performance on STS tasks (Spearman's correlation). SimCSE versus SimCSE with Deep Augmentation, specifically layer-targeted dropout and stop-gradient at layer 8. Compute refers to the estimated use of compute time and memory, as compared to SimCSE.

| Model | STS12 | STS13 | STS14 | STS15 | STS16 | STS-B | SICK-R | Avg. | Compute |
|---|---|---|---|---|---|---|---|---|---|
| SimCSE | 66.59 | 81.05 | 73.82 | 81.08 | **79.05** | 77.55 | 71.91 | 75.86 | 100% |
| SimCSE+DeepAug | **70.35** | **81.66** | **74.11** | **82.13** | 78.20 | **78.59** | **72.03** | **76.72** | ~79% |

**Compute & Memory Savings.** Stop-gradient reduces computational time and memory. In our setup, networks with gradient computations take roughly $4\times$ as much time and $3\times$ more memory than those without. Applying stop-gradient to higher layers and to half of the batch cuts computational time to 62.5% and memory to 66%. We present a rough estimate of the average computational savings in Tables 2, 3, and 4 under "Compute."

Table 2: Contrastive Learning on Sentence Embeddings with Transformer. Performance on STS tasks (Spearman's correlation). SimCSE versus SimCSE with Deep Augmentation, specifically layer-targeted dropout and stop-gradient at layer 8. Compute refers to the estimated use of compute time and memory, as compared to SimCSE.

| Model | STS12 | STS13 | STS14 | STS15 | STS16 | STS-B | SICK-R | Avg. | Compute |
|---|---|---|---|---|---|---|---|---|---|
| SimCSE | $66.71_{\pm0.505}$ | $81.13_{\pm1.279}$ | $73.13_{\pm1.818}$ | $80.82_{\pm0.593}$ | $\mathbf{78.47}_{\pm0.644}$ | $77.54_{\pm0.906}$ | $71.49_{\pm0.904}$ | $75.61_{\pm0.924}$ | 100% |
| SimCSE+DeepAug | $\mathbf{69.00}_{\pm1.111}$ | $\mathbf{81.82}_{\pm0.127}$ | $\mathbf{74.48}_{\pm0.311}$ | $\mathbf{81.84}_{\pm0.439}$ | $78.41_{\pm0.146}$ | $\mathbf{78.63}_{\pm0.114}$ | $\mathbf{71.75}_{\pm0.442}$ | $\mathbf{76.56}_{\pm0.161}$ | $\sim$79% |

Table 3: Contrastive Learning in Vision with ResNets. SimCLR versus SimCLR with Deep Augmentation, specifically layer-targeted dropout and stop-gradient at layer 4, across all datasets. Compute refers to the estimated use of compute time and memory, as compared to SimCLR.

| Model | CIFAR10 | CIFAR100 | ImageNet100 | Compute |
|---|---|---|---|---|
| SimCLR | 90.37 | 61.64 | 79.38 | 100% |
| SimCLR+DeepAug | **91.04** | **64.01** | 79.56 | $\sim$66% |

Table 4: Contrastive Learning on Graphs with GNNs. GCL (Graph Contrastive Learning) versus GCL with Deep Augmentation, specifically layer-targeted dropout and stop-gradient at layer 6, across all datasets. Compute refers to the estimated use of compute time and memory, as compared to GCL.

| Model | COLLAB | IMDB-Multi | NCI1 | PROTEINS | Compute |
|---|---|---|---|---|---|
| GCL | 72.40 | **53.33** | 73.97 | 72.32 | 100% |
| GCL+DeepAug | **73.80** | 52.88 | **75.83** | **73.21** | $\sim$66% |

## 5 Ablations

While Deep Augmentation introduces extra hyperparameters, we demonstrate consistent performance across datasets. This section presents ablation studies, highlighting performance variations among augmentations and targeted layers, and contrasts between contrastive and supervised learning.

### 5.1 Sentence Embeddings

SimCSE shows that using only dropout as an augmentation improves contrastive learning for sentence embeddings with a pre-trained MLM model. We examine how applying Deep Augmentation, via layer-specific dropout or PCA with or without stop-gradient, translates to the SimCSE setup under conditions with and without MLM's data augmentations.

**Augmentation, Layer, and Stop-Gradient.** Figures 2a and 2b display the effects of Deep Augmentation using dropout and PCA on the development set, resp. PCA augmentation was conducted with both the largest and sixth largest principal components; the former, yielding better outcomes, is in Figure 2b; see the Appendix for the latter.

These experiments demonstrate that Deep Augmentation improves performance across layers. Stop-gradient, when used with dropout, slightly enhances results, but its effectiveness is less pronounced with PCA.

**Deep Augmentation and Masked Language Modeling (MLM).** We assess Deep Augmentation alongside MLM's original data augmentation techniques, following SimCSE's experimental setup in Figure 3. Despite SimCSE's performance optimization across dropout rates (0%, 1%, 5%, 10%, 15%, 20%), it underperforms. Integrating Deep Augmentation with a standard 50% dropout rate boosts performance, highlighting the effectiveness of layer-specific augmentation. This combination enhances training robustness, reduces development set dependence, and supports simultaneous Deep Augmentation and MLM training. On STS tasks, the best Deep Augmentation setups outperform SimCSE's highest Spearman's correlation scores: 74.32 vs. 69.31. In the Appendix, we show that even for MLM without contrastive learning, Deep Augmentation markedly improves performance, and there stop-gradient's effect is less pronounced.

**Supervised Learning.** This section evaluates Deep Augmentation in supervised learning and compares it to contrastive learning outcomes. Deep Augmentation results with dropout and PCA are in Figures 4a and 4b, resp. In supervised learning, Deep Augmentation reduces performance, especially in higher layers, contrasting with the positive effects in contrastive learning.

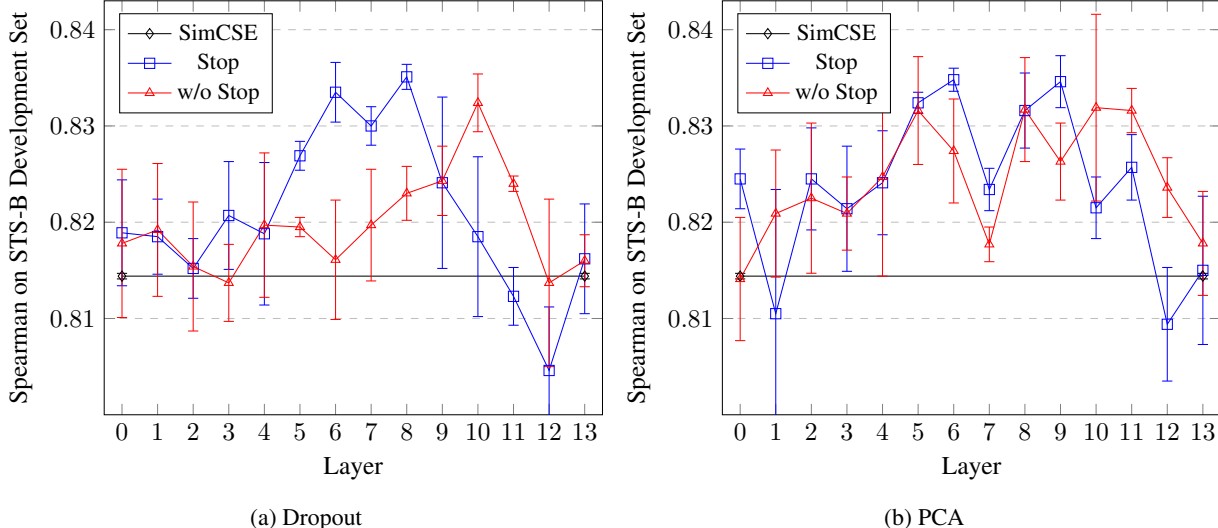

(a) Dropout  (b) PCA

Figure 2: SimCSE vs. Deep Augmentation with (a) Dropout or (b) PCA, with and without stop-gradient. "Stop": stop-gradient. Deep Augmentation outperforms SimCSE.

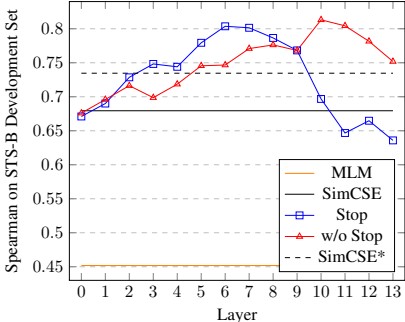

Figure 3: SimCSE vs. Deep Augmentation with and without stop-gradient, both with MLM. "Stop": stop-gradient. *: includes hyperparameter search over dropout rates.

## 5.2 Vision

Our primary results focus on the CIFAR-100 dataset, with corresponding outcomes for CIFAR-10 in the Appendix. For ImageNet100, we applied the optimal configuration from CIFAR, involving layer-targeted dropout and stop-gradient at the fourth layer.

**Architecture.** We use ResNet18 (Appendix A.1, Table 5). In contrast to transformers and GNNs, ResNet exhibits less regularity. Specifically, it comprises convolutional layers at the beginning, followed by a fully connected layer, with average pooling interposed, and the layers vary in dimensions.

**Dropout is not a Sufficient Augmentation.** In our experiments, using only dropout for augmentation, without other data augmentations, did not produce competitive outcomes. Thus, for vision, we combined "deep" augmentations (including dropout) with data augmentations. Future research could explore if using a development set for early stopping and hyperparameter selection enables adaptation of pre-trained vision models to new image data using dropout alone, similar to methods in SimCSE and Section 5.1.

**Data Augmentation and Targeted Dropout.** Our results show that general dropout reduces contrastive learning effectiveness. However, layer-specific application of dropout shows varying performance impacts across layers (Figure 5). Notably, a uniform 50% dropout rate significantly hinders performance, whereas targeting 50% dropout at certain layers results in a much smaller decrease in performance.

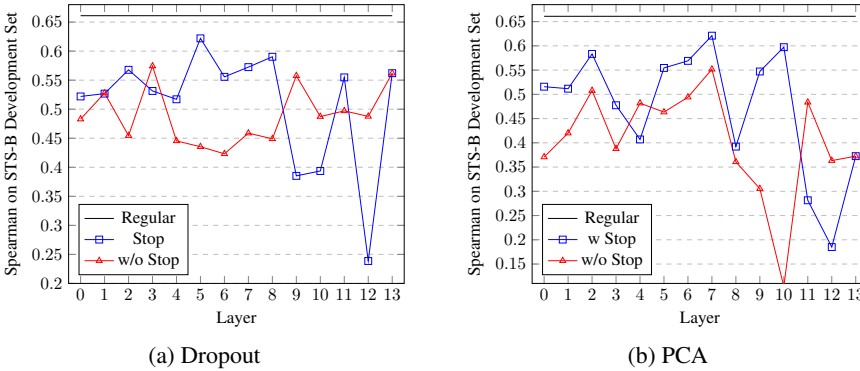

(a) Dropout              (b) PCA

Figure 4: Supervised only. Deep Augmentation with (a) Dropout (across dropout rates .5, .25, .125) and (b) PCA, with and witout stop-gradient, on STS-B.

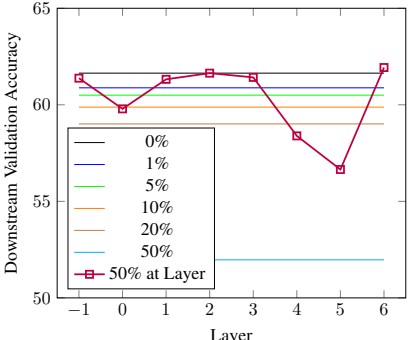

Figure 5: Comparing dropout rates at all layers versus 50% dropout rate targeted at a specific layer. For ratio of dropped to total nodes when targeting a layer, see Appendix A.2; there is no trend.

**Augmentation, Layer, and Stop-Gradient.** Now we apply Deep Augmentation with dropout (Figure 6a), with and without stop-gradient. Deep Augmentation with dropout and stop-gradient demonstrate significant performance improvements, particularly for layers 4 and 6; however, not using the stop-gradient did not achieve performance comparable to using stop-gradient. A small tuning of dropout rate yielded the results in Table 3 (Appendix A). We also evaluate Deep Augmentation with PCA augmentation, removing the largest and the sixth largest principal component. Removing the sixth largest yields superior performance (results with and without stop-gradient in Figure 6b). For the largest, see the Appendix. Similar to dropout, stop-gradient consistently enhances performance, especially in higher layers.

**Initialization & Freezing Weights.** Augmentations may have a more significant impact on higher layers that already possess useful, discriminative features. In addition, the concurrent objectives of learning features and maintaining invariance to their alterations could conflict, slowing down or destabilizing training. In Figure 6a, 'SimCLR*' and 'Stop*' show results using SimCLR and Deep Augmentation with stop-gradient (resp.), starting with weights from a SimCLR pre-trained network on CIFAR100. Despite Deep Augmentation showing a marginal superiority over SimCLR in terms of initialization, the advantages of pre-training before applying Deep Augmentation are minimal. This indicates that pre-training a neural network is not crucial for Deep Augmentation. Further, freezing layers before or after the targeted layer, with pre-trained SimCLR weights on CIFAR, did not yield competitive performance; see Appendix A.6.

**Supervised Learning.** We investigate Deep Augmentation in the context of supervised learning, comparing its efficacy to that in contrastive learning (Figures 7a and 7b). Deep Augmentation, whether through dropout or PCA, does not enhance performance in supervised learning on the CIFAR dataset. Notably, the absence of stop-gradient yielded superior results compared to its inclusion, diverging from the trend in contrastive learning. While Deep Augmentation appears to detract from performance in supervised learning, retaining data augmentations is crucial, as their omission resulted in a significant drop in accuracy to 59.02%.

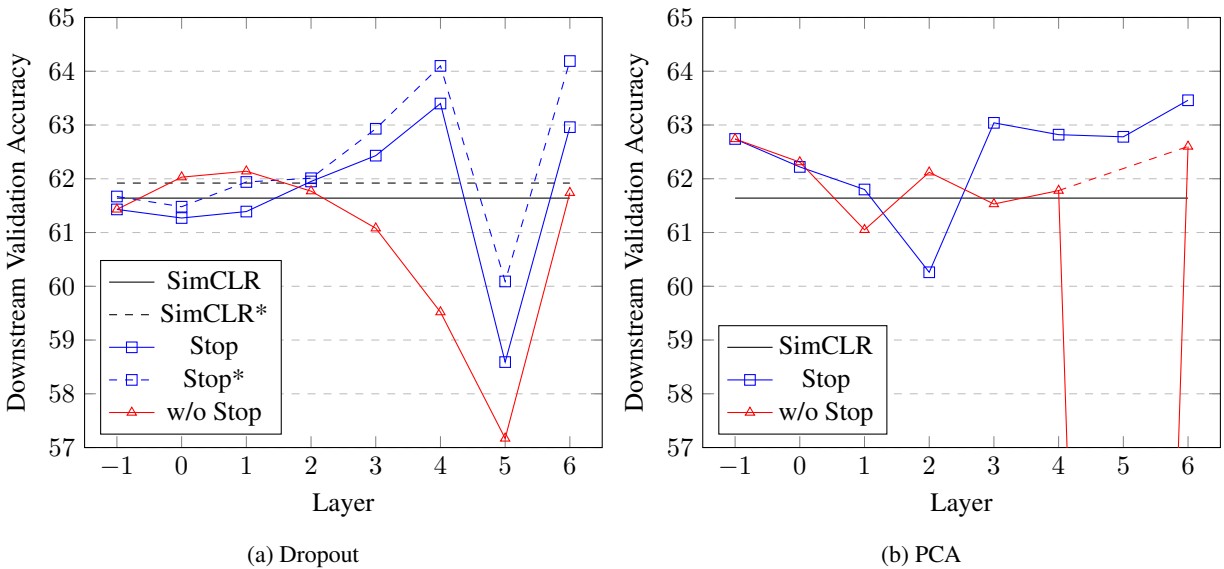

Figure 6: Contrastive learning. Deep Augmentation with (a) dropout or (b) PCA, with and without stop-gradient. *: initialized with pre-trained SimCLR model. "Stop" is short for stop-gradient. Note: Layer 5 is an average pooling.

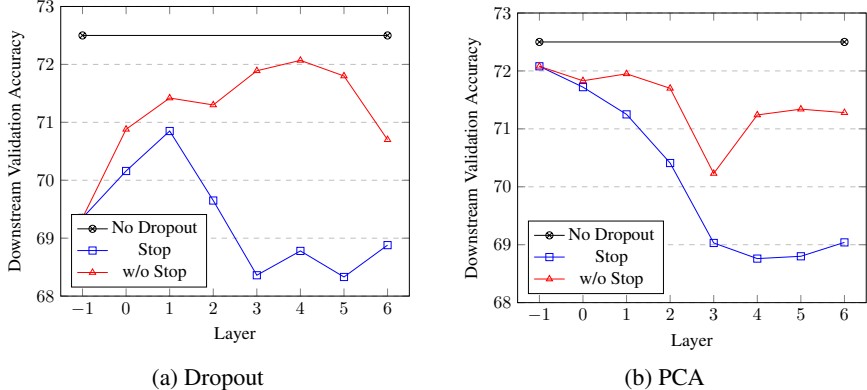

Figure 7: Supervised only. Deep Augmentation with (a) dropout or (b) PCA, with and without stop-gradient. *: initialized with pre-trained SimCLR model. "Stop" is short for stop-gradient.

## 5.3 Graphs

Next, we explore transferability of Deep Augmentation insights from vision and sentence embeddings to graph contrastive learning, assessing applicability across modalities.

**Augmentation, Layer, and Stop-Gradient.** We apply Deep Augmentation to graph datasets including COLLAB, IMDB-Multi, NCI1, and PROTEINS, integrating it with standard graph data augmentation techniques. The results of Deep Augmentation using dropout and PCA (targeting the 6th largest principal component) are presented in Figures 8 and 9, respectively. Notably, due to the variable embedding size of each graph neural network (GNN) sample within a batch, we adapted the PCA to operate over all node embeddings.

While trends across these datasets are not uniform, Deep Augmentation, particularly when with dropout and stop-gradient, significantly enhanced performance on all datasets.

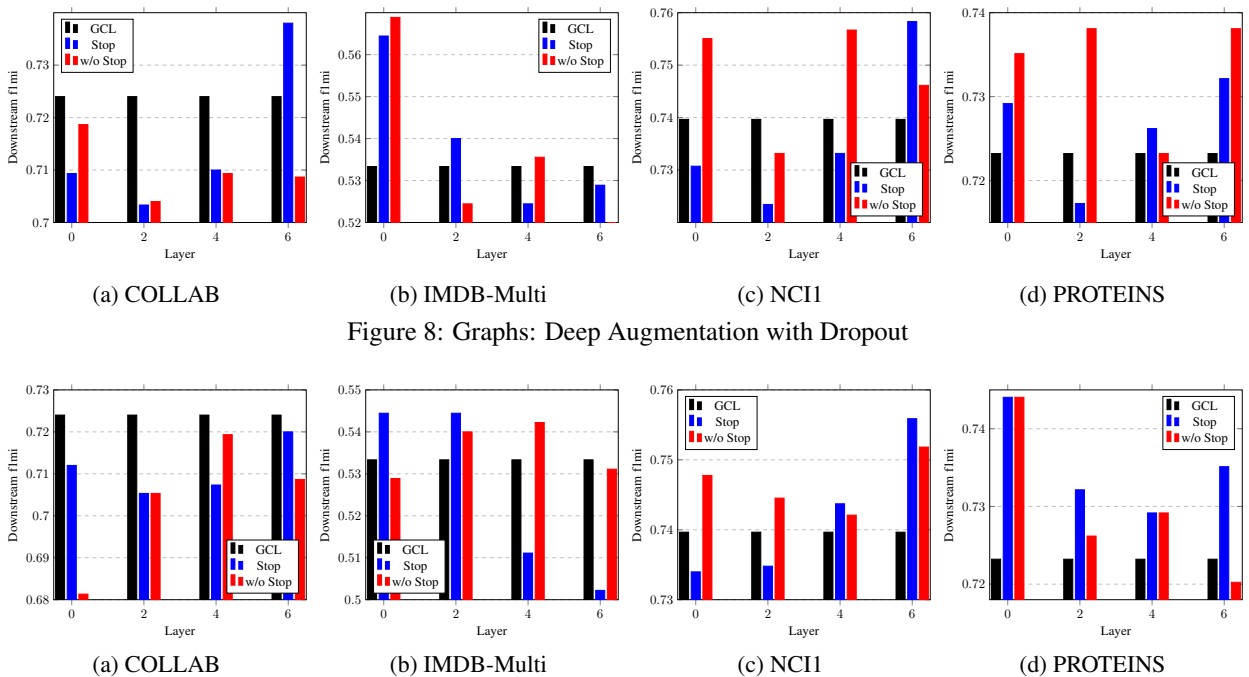

Figure 8: Graphs: Deep Augmentation with Dropout

Figure 9: Graphs: Deep Augmentation with PCA

# 6 Analysis

Our analysis suggests that Deep Augmentation aids contrastive learning by reducing overfitting and eliminating spurious alignment, while maintaining or enhancing uniformity. This is evident in uniformity and alignment measures and corroborated by CKA analysis, which shows that Deep Augmentation decreases similarity between network layers through stronger feature transformations, referred to as reduced co-adaptation between layers. Additionally, the CKA analysis identifies which layers are most susceptible to co-adaptation and where Deep Augmentation is most effective.

In contrast, our analysis indicates that Deep Augmentation does not benefit supervised learning because the ground truth labels inherently counteract the spurious alignment present in contrastive learning. In supervised learning, the task is fundamentally different, with ground truth invariances already specified, whereas in contrastive learning, we optimize mutual information across an infinite number of augmentations and invariances. This results in supervised learning exhibiting less co-adaptation between layers. This observation may be analogous to the information bottlenecks seen in supervised versus self-supervised settings (Tishby & Zaslavsky, 2015; Shwartz-Ziv & LeCun, 2023; Jing et al., 2022).

## 6.1 Co-adaptation between Layers

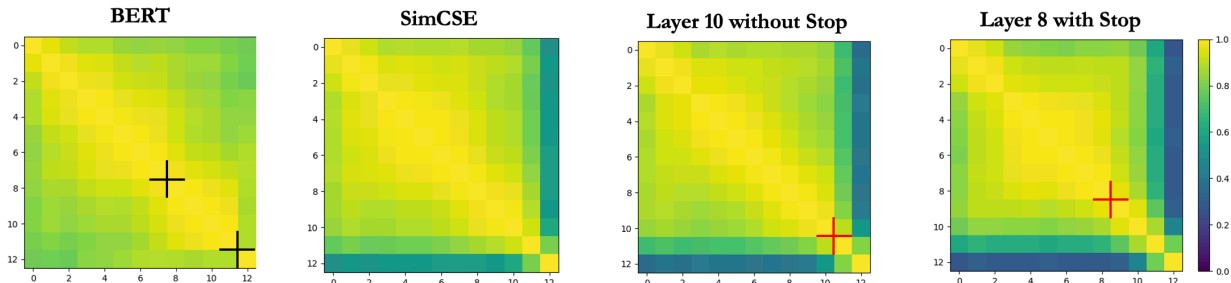

Figure 10: CKA similarity index for "BERT," "SimCSE," "Layer 10 without Stop" (Deep Augmentation applied without stop-gradient at Layer 10), and "Layer 8 with Stop" (Deep Augmentation applied with stop-gradient at Layer 8) on the STS-B dataset. Black crosses mark the beginning and end of a co-adaptation region in BERT, while red crosses on "Layer 10 without Stop" and "Layer 8 with Stop" highlight the targets of Deep Augmentation. Optimal performance of Deep Augmentation is observed near the black crosses, indicating its effectiveness and guiding the selection of layers for targeting.

**Sentence Embeddings and Transformer.** We compute the CKA similarity index for the Transformer trained under various settings (Figure 10). "BERT" is the starting point for both SimCSE and Deep Augmentation."BERT" has a stretch of co-adaptation between layers 8 through 11 as indicated by the two black crosses. Mirroring ResNet18 on CIFAR, it is around these two crosses that Deep Augmentation is most effective, with stop-gradient at the earlier cross and without stop-gradient at the later cross. "SimCSE" reduces the co-adaptation across layers, especially between layers 11-10-12. In "Layer 10 without Stop" and "Layer 8 with Stop" the red cross indicates where Deep Augmentation was applied. "Layer 10 without Stop" and "Layer 8 with Stop" supersede SimCSE in reducing co-adaptation among the layers following their application. Thus, CKA similarity index highlights a co-adaptation issue between layers and determines at what layer Deep Augmentation should be applied. Future work may investigate if simultaneously targeting multiple layers further improves performance.

In Figure 57 in the Appendix, we note that the supervised setting exhibits significantly reduced co-adaptation even in the absence of Deep Augmentation, indicating this issue is less pronounced in such contexts. Further details are provided in Appendix E.

**Images and ResNet.** Figure 11 shows CKA for a randomly-initialized ResNet18, after training with SimCLR and after trainining with Deep Augmentation without stop-gradient post-Layer 4. We include results for more settings in the Appendix, but most resemble "SimCLR" or "Layer 4 without Stop." There is strong co-adaptation between Layer 4 and 5 after training with SimCLR but not before, and poorly performing "Layer 4 without Stop" has increased co-adaptation between Layer 4 and 5. In contrast, top performing Layer 4 and 6 are similar to "SimCLR." This suggests a failure case involving increased co-adaptation between layers, and that Layer 4 is special because ResNet18 is susceptible to co-adaptation between Layer 4 and subsequent layers. This also pinpoints where Deep Augmentation is most effective, a pattern similarly noted in our sentence embeddings analysis.

In Figure 54 in the Appendix, we observe that the supervised setting exhibits reduced co-adaptation compared to the self-supervised settings, particularly in the last layer. This consistent difference, as also observed with sentence embeddings, underscores a disparity between self-supervised and supervised learning.

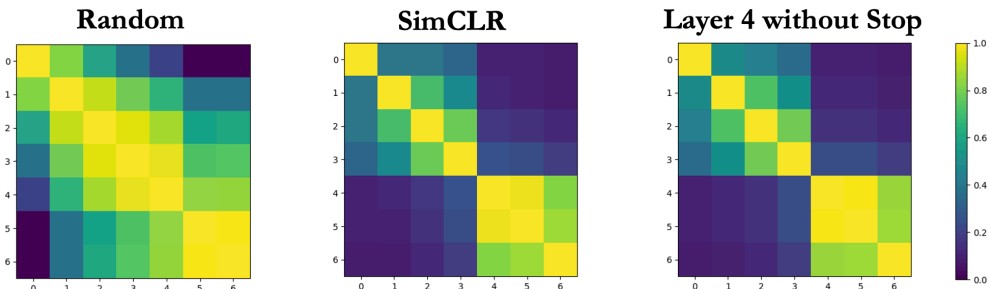

Figure 11: Indications of why Layer 4 is special in Figure 6a, as the major divide between co-adaptation across layers. Layers 0-4 are convolutional. All high-performing NNs have the pattern of SimCLR, and failure cases have stronger co-adaptation between Layers 4 and 5. "Layer 4 without Stop" corresponds to the failure case of Deep Augmentation without stop-gradient at Layer 4.

## 6.2 Alignment and Uniformity

**Sentence Embeddings and Transformer.** Alignment and Uniformity measures for sentence embedding methods are in Figure 12, computed as in SimCSE (Gao et al., 2021) w.r.t. ground truth (STS-B development set), during training, and with methods converging to higher density regions.

Without MLM, all methods converge toward better Uniformity at the expense of Alignment. "S" is consistently better in both measures than "SimCSE," while "w/o S" further encourages Uniformity at the expense of Alignment. With MLM, the direction reverses, and all methods improve Alignment. Again "S+mlm" (MLM and Deep Augmentation with stop-gradient) is consistently better than "SimCSE+mlm" in both Alignment and Uniformity. "w/o S+mlm" (MLM and Deep Augmentation without stop-gradient) achieves markedly improved Uniformity compared to "SimCSE+mlm," albeit at the cost of Alignment.

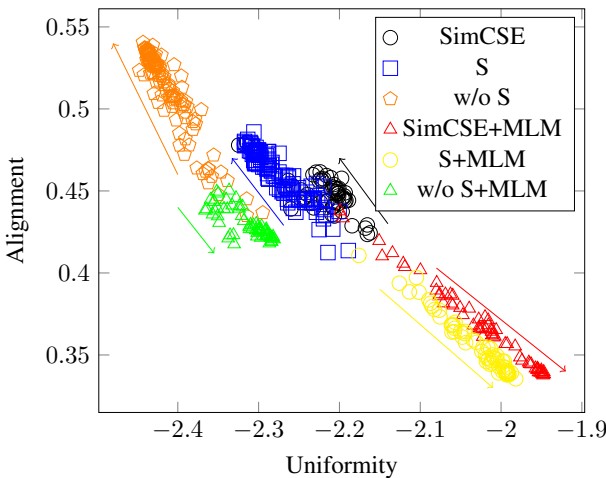

Figure 12: Alignment and Uniformity (lower is better) for sentence embeddings on STS-B: SimCSE vs. Deep Augmentation (with and without stop-gradient). We also include these methods combined with the pre-training method of BERT, i.e., Masked Language Modeling (MLM). Arrows indicate the direction during training, which reverses when MLM is introduced. "S" is short for stop-gradient.

This suggests that Deep Augmentation generally achieves better Alignment and Uniformity than SimCSE, particularly in terms of Uniformity. This analysis is consistent with that of SimCSE Gao et al. (2021). However, since these measures are computed on ground truth validation classes rather than augmentations, they offer a different perspective from the original introduction of these measures for contrastive learning Wang & Isola (2020).

In Table E in the Appendix, we present Alignment and Uniformity metrics for supervised learning on the same validation ground truth classes. The results indicate that Deep Augmentation achieves better Alignment but worse Uniformity. Notably, Uniformity is already higher for regular supervised learning compared to self-supervised approaches, and Deep Augmentation does not improve this metric.

**Images and ResNet.** The Alignment/Uniformity measures depend on specific augmentations. Our approach, involving different augmentations, complicates direct comparison, so we standardized using SimCLR's data augmentations (Figure 13). Analysis over five checkpoints (epochs 300, 600, 900, 1200, 1500) shows more training generally improves Uniformity on test data and Alignment on training data.

Layers 4 and 6 with stop-gradient outperform in Alignment and Uniformity using SimCLR augmentations on test data, and in Uniformity on the training data. Conversely, Layer 4 without stop-gradient and SimCLR exhibit better Alignment on training data, suggesting potential overfitting of Alignment as their test set Alignment degrades with training. Layer 5 performs poorly in both sets, aligning with downstream results. Despite differences in downstream tasks, SimCLR and Layer 4 without stop-gradient have similar Alignment and Uniformity, highlighting these metrics' limitations in capturing latent space quality. Hence, our preference for the CKA similarity index for comprehensive assessment.

In Figure 52 in the Appendix, we present the Alignment and Uniformity metrics for supervised models trained on ground truth labels. While no overfitting is observed, Layers 4 with stop-gradient again perform better on both metrics. This highlights that the invariances in self-supervised learning and supervised learning remain fundamentally different, and superior performance in the self-supervised task does not necessarily translate to improved performance in the supervised downstream task. Avoiding overfitting to Alignment is crucial when that Alignment differs from the ground truth, but less critical when Alignment matches the ground truth.

## 7 Discussion

This paper presents the multifaceted impacts of Deep Augmentation techniques on contrastive and supervised learning. We demonstrate the efficacy of layer-targeted dropout, both with and without stop-gradient, in enhancing contrastive learning across modalities. This effect is most pronounced in higher layers, underscoring the importance of strategic

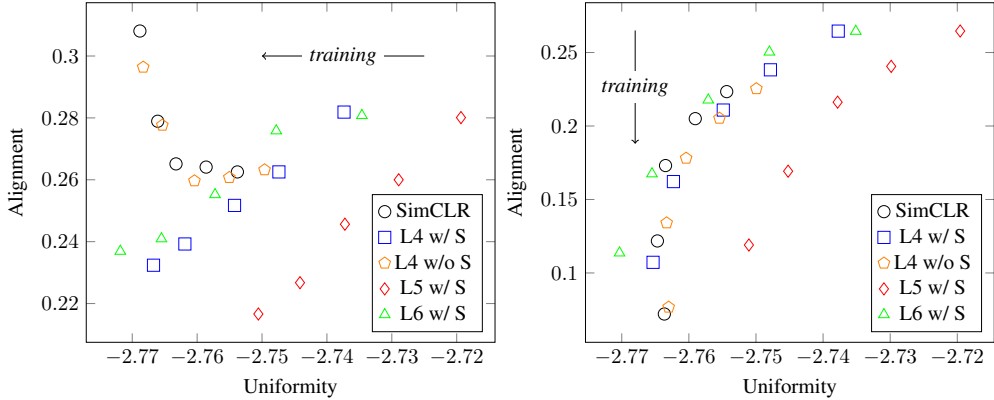

Figure 13: Alignment and Uniformity (lower is better) of SimCLR augmentations on CIFAR. Left: Test data. Right: Training data. Deep Augmentation outperforms SimCLR when measuring alignment and uniformity *using SimCLR's augmentations* on the test set, and SimCLR overfits at Alignment on the training set. "L" is short for Layer and "S" is short for stop-gradient.

layer selection in augmentation techniques. Contrastingly, our investigation into supervised learning contexts reveals an inverse relationship.

Furthermore, we introduce another modality-agnostic PCA-based augmentation strategy, demonstrating its use as an alternative to dropout. This approach not only mirrors performance enhancements attributed to dropout but also provides a versatile tool applicable across learning scenarios.

Our findings address the issue of co-adaptation among network layers and their latent features. Our targeting of specific layers underscores the potential of informed, strategic interventions in network training. While much future work is needed to fully understand the mechanisms of self-supervised learning, our analysis indicates that Deep Augmentation aids contrastive learning by reducing overfitting and eliminating spurious alignments, while maintaining or even enhancing uniformity.

Extensive ablation studies underpin our contributions, providing a robust empirical foundation. These pave the way for future research to explore the interplay between augmentation techniques, network architectures, and learning paradigms, with the ultimate goal of enhancing machine learning models' efficiency and versatility.

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

# A  CIFAR

In this section, we outline the training specifics for our experiments on the CIFAR datasets, complemented by supplementary results and comparative analyses.

## A.1  ResNet architecture

The specifications for ResNet18 are detailed in Table 5.

Table 5: Configuration of ResNet18 on CIFAR

| ResNet18 on CIFAR | | |
|---|---|---|
| Layer | Type | #Neurons |
| -1 | Input Data | $32^2 \times 3 = 3072$ |
| 0 | Conv(k=3, s=1) | $32^2 \times 64 = 65536$ |
| 1 | Conv(k=3, s=2) | $32^2 \times 64 = 65536$ |
| 2 | Conv(k=3, s=2) | $16^2 \times 128 = 32768$ |
| 3 | Conv(k=3, s=2) | $8^2 \times 256 = 16384$ |
| 4 | Conv(k=3, s=2) | $4^2 \times 512 = 8192$ |
| 5 | Avgpool | 512 |
| 6 | MLP | 128 |

## A.2  Proportion of Dropped Nodes

When setting a dropout rate for a specific layer, it exclusively affects that layer. Consequently, a 50% dropout rate at one layer results in fewer neurons being dropped compared to a 50% dropout applied uniformly across all layers. Additionally, the same dropout rate can impact different numbers of neurons in various layers, reflecting the varying neuron counts in each layer.

In Table 6 we include the number of nodes in each layer, the total nodes across all layers. Thus, for $0.5$ dropout, we show the proportion of dropped nodes when a layer is targeted. There is not trend between the proportion and performance.

Table 6: Proportion of Dropped Nodes per Layer at 50% dropout

| Layer | Dropped Nodes | Total Nodes | Proportion |
|---|---|---|---|
| 0 | $0.5 \times 65536$ | 192640 | 0.17 |
| 1 | $0.5 \times 65536$ | 192640 | 0.17 |
| 2 | $0.5 \times 32768$ | 192640 | 0.085 |
| 3 | $0.5 \times 16384$ | 192640 | 0.043 |
| 4 | $0.5 \times 8192$ | 192640 | 0.021 |
| 5 | $0.5 \times 512$ | 192640 | 0.001 |
| 6 | $0.5 \times 128$ | 192640 | 0.0003 |

## A.3  Training Details

For implementation, we utilized the code provided by (Khosla et al., 2020), available at this link. Our experiments were conducted with a batch size of 1024, training each method for 1500 epochs.

## A.4  Naïve Deep Augmentation with stop-gradient on CIFAR100

In Figure 14, we include results of 50% dropout with stop-gradient at individual layers on 100% of the batch. Such naïve augmentations generally give poor performances. All layers besides the input data layer lead to downstream accuracy of 1% (equivalent with random guess). The input data layer arrives at a downstream accuracy of 61.38%.

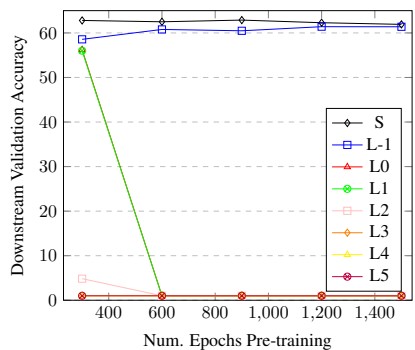

Figure 14: CIFAR100. 50% dropout with stop-gradient applied at individual layers on 100% of the batch. I.e. freezing earlier layers to random weights.

## A.5 Including Deep Augmentation w/o stop gradient initialized with SimCLR

For completion, we also include Deep Augmentation without stop gradient, initialized with pre-trained SimCLR model, together with the other variants—see Figure 15.

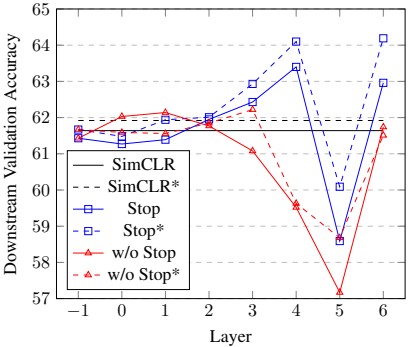

Figure 15: Comparing sampling 50% and applying 50% dropout, with or without stop-gradient. *: initialized with pre-trained SimCLR model. Stop: short for stop-gradient.

## A.6 Freezing

We repeat the experiment with pre-trained initialization but freeze all the layers up to and including the layer at which the targeted transformation occurs; see "Freeze before" in Figure 16. Compared to not freezing, this strategy gives very different results. In particular, the downstream performance of Layers 3 and 4 is critically reduced.

Deep Augmentation after frozen SimCLR layers may not work well due to co-adaptation between neurons, leading to overfitting. Suppose a layer of a NN exhibits strong co-adaptation within several subsets of neurons, i.e., each subset encodes a single data feature. Randomly dropping neurons is unlikely to remove a complete co-adapted subset of neurons. Ideally, features are learned per neuron so dropping any of them provides a complementary view. Alternatively, features might be represented continuously among neurons in a layer such that dropout corresponds to something akin to blurring the feature continuously.

Because early layers have fewer parameters to distort the input data, such layers may have less co-adaptation. This might explain why earlier layers, rather than later layers, perform better when frozen during Deep Augmentation. Similarly, higher layers may benefit from higher dropout rates because they are more susceptible to co-adaptation, explaining why in Figure 6a, Deep Augmentation in higher layers yields the best downstream performance.

Reversely, we may freeze the layers following the targeted layer; results are labeled "Freeze after" in Figure 16. Compared to "Freeze before", Layer 3 improves, Layer 5 worsens, while Layer 4 performs similarly. This asserts that later layers, some more than others, benefit from learning to be invariant to Deep Augmentation.

We see that Deep Augmentation with freezing layers and initialized to SimCLR-model, works better for earlier layers than for later layers. Especially in Figure 38a and 42, we see that earlier layers outperform SimCLR earlier in the training. This suggests that incrementally freezing layers, and adding Deep Augmentation at the next layer, might help improve performance and speed up training.

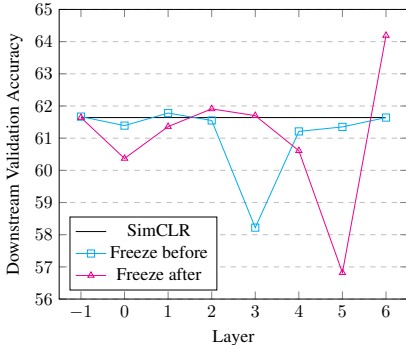

Figure 16: Freezing layers before and after Deep Augmentation with stop-gradient, initialized with pre-trained SimCLR model. For "Freeze before," Layer -1 freezes nothing, and for "Freeze after" Layer 6 freezes nothing.

## A.7 PCA Augmentation

In Figure 20, results demonstrate that removing the largest principal component from a batch sample is less effective than subtracting the sixth largest (Figure 27).

Figure 19 presents the six largest principal values from the layers of a randomly initialized ResNet18 versus one trained with SimCLR on CIFAR100. Post-SimCLR training, the distribution of values becomes more uniform, and there is a notable shift in the rank of layers before and after the training process.

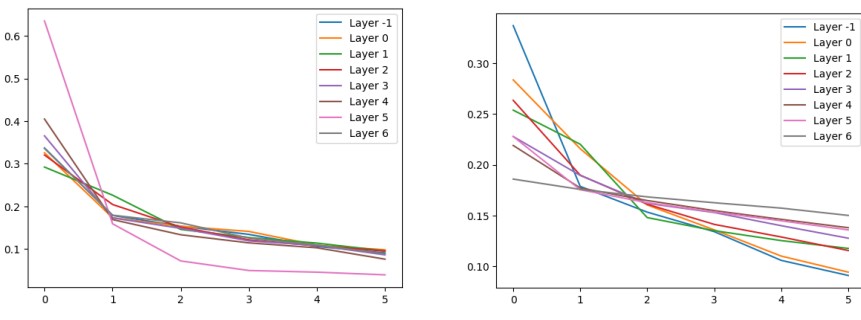

Figure 17: Random init.               Figure 18: After SimCLR

Figure 19: The six largest principal value from the layers of a random initialized ResNet18 and one trained with SimCLR on CIFAR100.

## A.8 Supervised Learning

For our supervised learning experiments, training was conducted for 100 epochs but otherwise using the same hyperparameters as those in the fine-tuning phase post pre-training, which lasted 28 epochs.

Figure 22 presents results from supervised learning on CIFAR100, comparing the effects of uniform dropout across all layers with 50% dropout applied to a targeted layer.

Figure 21 presents the results of supervised training but also includes standard deviations.

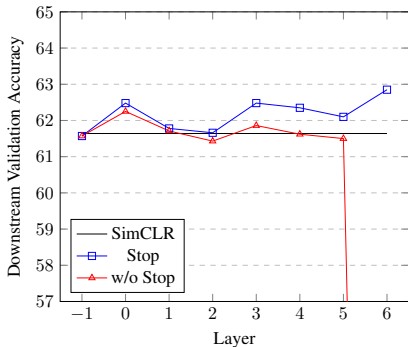

Figure 20: PCA: Comparing sampling 50% of batch and subtracting the largest principal component from that sample, with and without stop-gradient.

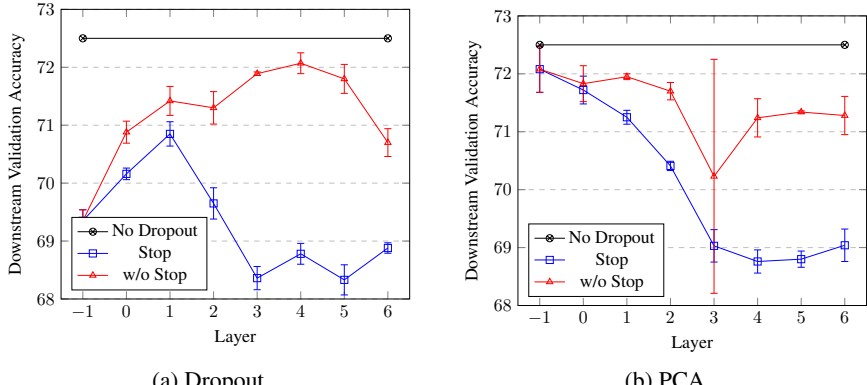

(a) Dropout             (b) PCA

Figure 21: Supervised only. Deep Augmentation with (a) dropout or (b) PCA, with and without stop-gradient. *: initialized with pre-trained SimCLR model. "Stop" is short for stop-gradient.

## A.9 Domain Transfer: CIFAR100 to CIFAR10

We perform basic domain transfer experiments by taking networks pretrained on CIFAR100 and finetuning them on CIFAR10. In Figure 23 we include results comparing SimCLR with Deep Augmentation with and without stop-gradient, across layers. We also include performance for different checkpoints across training, see Figure 24a and Figure 24b for Deep Augmentation with and without stop gradient, respectively. Note the overfitting tendencies.

## A.10 Different dropout rates

In Figure 25, we tune over dropout rates 0.5, 0.25, and 0.125 and find that 0.125 at Layer 4 performs the best.

## A.11 CIFAR10

We include results on most of the experiments that were run on CIFAR100, also on CIFAR10. In general, results show the same trends as for CIFAR100. In Figure 29, we include results comparing dropout rates across all layers to 50% dropout at single layers. Again, we see targeted dropout at some layers showing much better performance than dropout across all layers.

In Figure 30 we include results of sampling 50% of batch and performing 50% dropout with and without stop-gradient, called "Stop" and "w/o Stop" respectively. We also include a benchmark of SimCLR. Here "*" refers to the networks being initialized with a pre-trained SimCLR model. Again, we see Layer 4 (with stop-gradient) and Layer 6 (with and without stop-gradient) stand out. It is also interesting to note that when initializing with a pre-trained SimCLR model, performance differs significantly more for Deep Augmentation with stop-gradient than without.

In Figures 26 and 27, we subtract the largest and sixth largest principal component from half the samples of the batch.

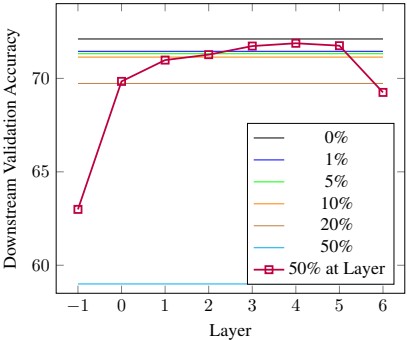

Figure 22: Supervised on CIFAR100: Comparing dropout rates at all layers versus 50% dropout rate targeted at a specific layer.

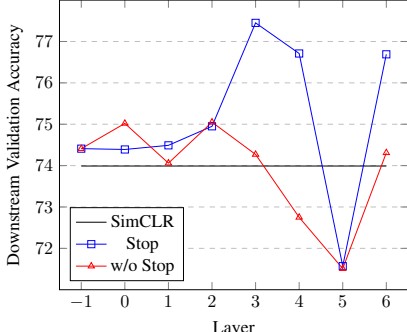

Figure 23: Finetuning on CIFAR10 of networks pre-trained on CIFAR100. Comparing SimCLR with Deep Augmentation with and without stop-gradient. Stop: short for stop-gradient.

In Figure 31, we include results of Deep Augmentation with stop-gradient but freezing layers up to the targeted layer versus freezing after the targeted layer. Again, we see the performance change, especially Layer 3 and 4 degrading, while Layer 2 improves.

In Figure 28, we include results of supervised learning on CIFAR10, with dropout across all layers as well as 50% dropout at targeted-layer.

We perform basic domain transfer experiments by taking networks pretrained on CIFAR10 and finetuning them on CIFAR100. In Figure 32 we include results comparing SimCLR with Deep Augmentation with and without stop-

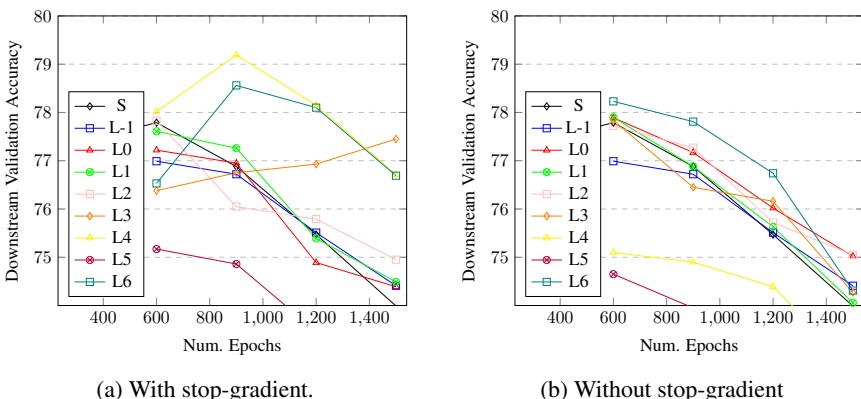

(a) With stop-gradient.            (b) Without stop-gradient

Figure 24: SimCLR and Deep Augmenation with and without stop-gradient pre-trained on CIFAR100 and finetuned on CIFAR10, for different checkpoints during training. Observe the overfitting behavior.

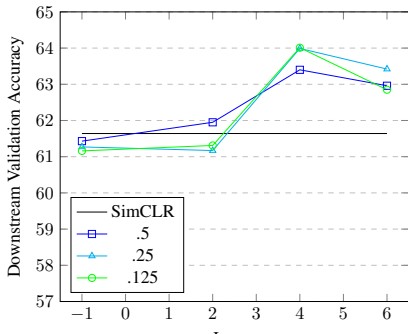

Figure 25: CIFAR100: Comparing sampling 50% of batch and applying dropout to that sample, with and without stop-gradient, for different dropout rates. "Stop" is short for stop-gradient.

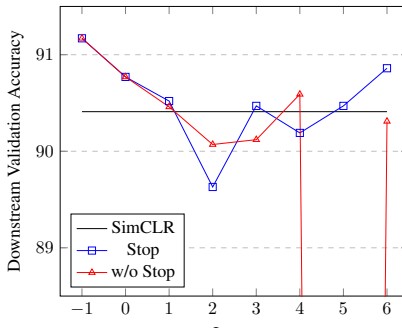

Figure 26: PCA CIFAR10: Comparing sampling 50% of batch and subtracting the largest principal component from that sample, with and without stop-gradient.

gradient, across layers. We also include performance for different checkpoints across training, see Figure 44a and Figure 44b for Deep Augmentation with and without stop gradient, respectively. Note the overfitting tendencies.

In Figure 33, we tune over dropout rates 0.5, 0.25, and 0.125 and fine that 0.125 at Layer 4 performs the best.

## A.12  CIFAR100 across epochs

We include results where we finetuned and tested checkpoints at different epochs for various experiments.

In Figure 34, we include results for dropout everywhere at different rates and 50% dropout at single layers.

In Figure 35, we include results for sampling 50% of each batch and performing 50% dropout on that sample, with and without stop-gradient.

In Figures 36 and 37, we include results for sampling 50% of each batch and subtracting the largest and sixth largest (respectively) principal component from that sample, with and without stop-gradient.

In Figure 38, we compare freezing layers before or after Deep Augmentation with stop-gradient initialized with pre-trained SimCLR model.

In Figure 39, we inlcude results for 50% sampling, 50% dropout, with and without stop-gradient, and initialized with pre-trained SimCLR model.

## A.13  CIFAR10 across epochs

We include results where we finetuned and tested checkpoints at different epochs for various experiments.

In Figure 40, we include results for dropout everywhere at different rates and 50% dropout at single layers.

In Figure 41, we include results for sampling 50% of each batch and performing 50% dropout on that sample, with and without stop-gradient.

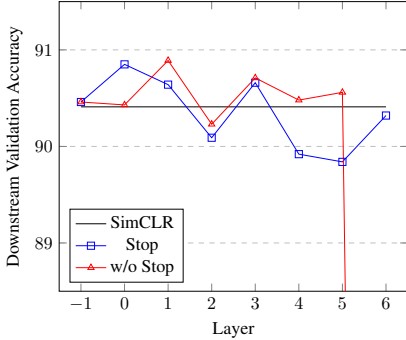

Figure 27: PCA CIFAR10: Comparing sampling 50% of batch and subtracting the sixth principal component from that sample, with and without stop-gradient.

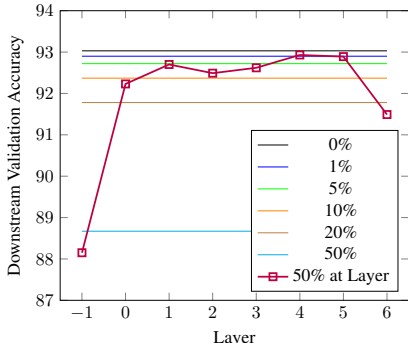

Figure 28: Supervised on CIFAR10: Comparing dropout rates at all layers versus 50% dropout rate targeted at a specific layer.

In Figure 42, we compare freezing layers before or after Deep Augmentation with stop-gradient initialized with pre-trained SimCLR model.

In Figure 43, we inlcude results for 50% sampling, 50% dropout, with and without stop-gradient, and initialized with pre-trained SimCLR model.

### A.14 CIFAR100 Miscellaneous Experiments

We include some preliminary results on different aspects of Deep Augmentation that deserve further investigation.

In Figure 45, we include results of Deep Augmentation with stop-gradient where each pair consists of one sample that has only input-data augmentation and another sample that has input-data and higher-layer augmentation. I.e. we remove all the higher-to-higher and lower-to-lower pairs. We see that for Layer 4 and 6 the performance does not change substantially, but for Layer 3 performance degrades substantially.

In Figure 46, we include results of Deep Augmentation without stop-gradient where each pair consists of one sample that has only input-data augmentation and another sample that has input-data and higher-layer augmentation. I.e. we remove all the higher-to-higher and lower-to-lower pairs. We see that for the layers involved performance does not change substantially.

This suggests that lower-to-higher pairs are sufficient to make Deep Augmentation successful, but that certain layers are greatly helped by also including other lower-to-lower or higher-to-higher pairs.

In Figure 47, we include results of Deep Augmentation with stop-gradient and freezing layers before, but initialized with random weights instead of initialized with a pre-trained SimCLR model. We note that several layers are severely hurt by this compared to the SimCLR pre-trained model initialization.

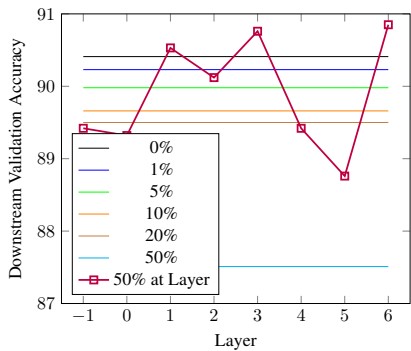

Figure 29: CIFAR10: Comparing dropout rates at all layers versus 50% dropout targeted at a specific layer.

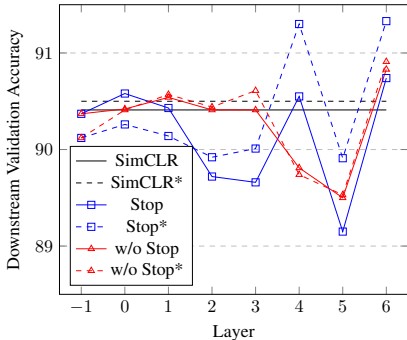

Figure 30: CIFAR10: Comparing SimCLR with Deep Augmentation with and without stop-gradient. *: Initialized with pre-trained SimCLR model. Stop: short for stop-gradient.

In Figure 48, we include results of Deep Augmentation with stop-gradient and freezing layers before, but initialized with a model pre-trained with SimCLR and 20% dropout across all layers. We wanted to see if a model trained with high dropout everywhere was more helpful as a starting point for Deep Augmentation. Future work may investigate ways to optimally train a NN so that dropout serves as a useful higher transformation.

# B  Sentence Embeddings

## B.1  Training Details

We used the training protocol of (Gao et al., 2021) with code released at link. Deep Augmentation at Layer 0 correspond to just after the first token-embeddings. Deep Augmentation at the subsequent layers was applied after each transformer layer in the code, with the last Layer 13 corresponding to the output latent vector.

## B.2  PCA Augmentation

In Figure 49, we include results with Deep Augmentation and subtracting the sixth largest principal component.

## B.3  Deep Augmentation and MLM only

In Figure 50, we include results with Deep Augmentation and MLM only, without contrastive learning. Deep Augmentation boosts performance substantially. This demonstrates that Deep Augmentation can boost performance in self-supervised learning settings beyond contrastive learning.

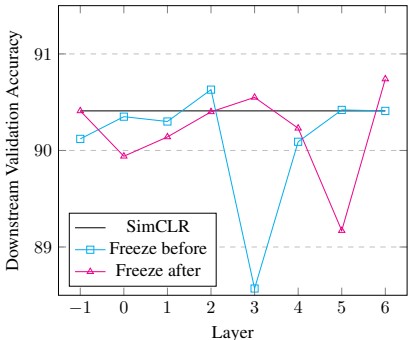

Figure 31: CIFAR10: Comparing freezing layers before or after Deep Augmentation with stop-gradient, initialized with pre-trained SimCLR model. Note that for "Freeze before" Layer -1 freezes nothing, and for "Freeze after" Layer 6 freezes nothing.

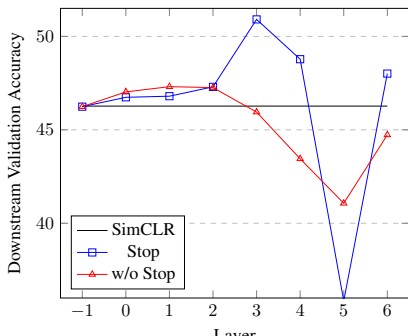

Figure 32: Finetuning on CIFAR100 of networks pre-trained on CIFAR10. Comparing SimCLR with Deep Augmentation with and without stop-gradient.

### B.4 Additional Results

In Figure 51, we include results of different dropout rates and hyper-parameter settings for using Deep Augmentation with SimCSE.

## C Graph Contrastive Learning

We follow the protocol and code of Zhu et al. (2021) that can be found at `https://github.com/PyGCL/PyGCL`. We pre-train for 1000 epochs and use the following data augmentations in GCL:

```
A.RandomChoice([
  A.RWSampling(num_seeds=1000,
    walk_length=10),
  A.NodeDropping(pn=0.1),
  A.FeatureMasking(pf=0.1),
  A.EdgeRemoving(pe=0.1)],
1)
```

For ablation study with standard deviations, see Table 7.

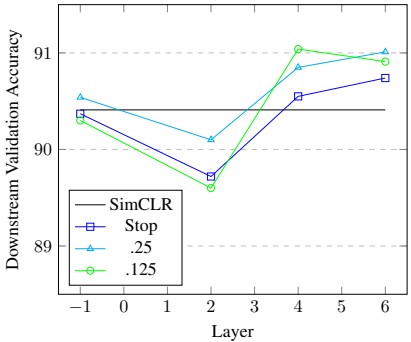

Figure 33: CIFAR10: Comparing sampling 50% of batch and applying dropout to that sample, with and without stop-gradient, for different dropout rates. "Stop" is short for stop-gradient.

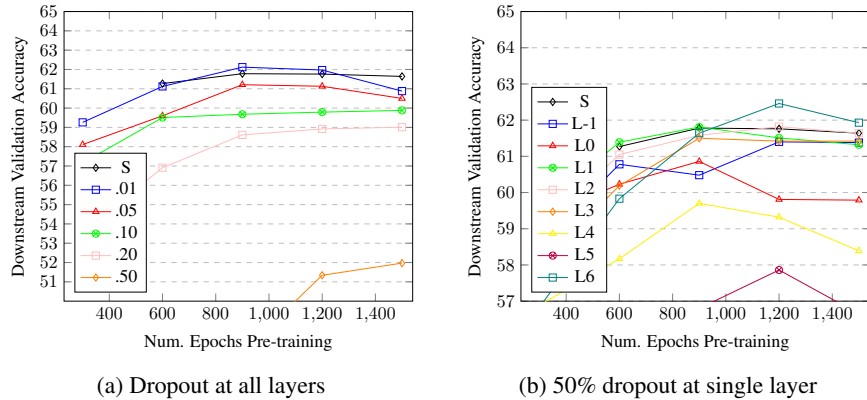

(a) Dropout at all layers

(b) 50% dropout at single layer

Figure 34: CIFAR100. Comparing dropout rates at all layers versus 50% dropout targeted at a specific layer. Note difference in $y$-axis.

# D  Alignment and Uniformity

First, we reiterate the fundamental drawbacks of Alignment and Uniformity for our work. Alignment is defined either with respect to a set of augmentations (the original intent Wang & Isola (2020)) or with respect to embeddings from different datapoints within the same class (as in SimCSE Gao et al. (2021)).

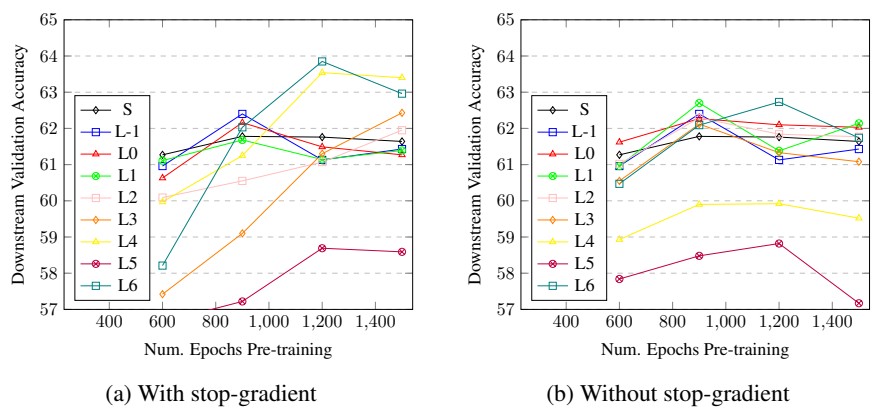

(a) With stop-gradient

(b) Without stop-gradient

Figure 35: CIFAR100. Comparing sampling 50% and applying 50% dropout, with or without stop-gradient.

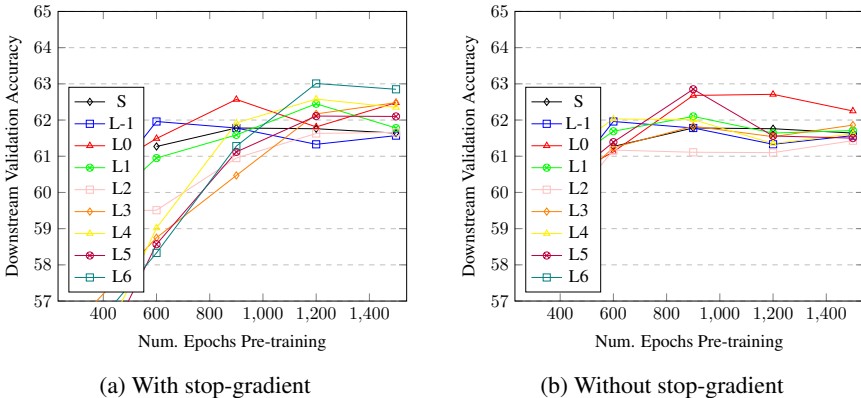

(a) With stop-gradient  (b) Without stop-gradient

Figure 36: PCA CIFAR100: Comparing sampling 50% of batch and subtracting the largest principal component from that sample, with and without stop-gradient.

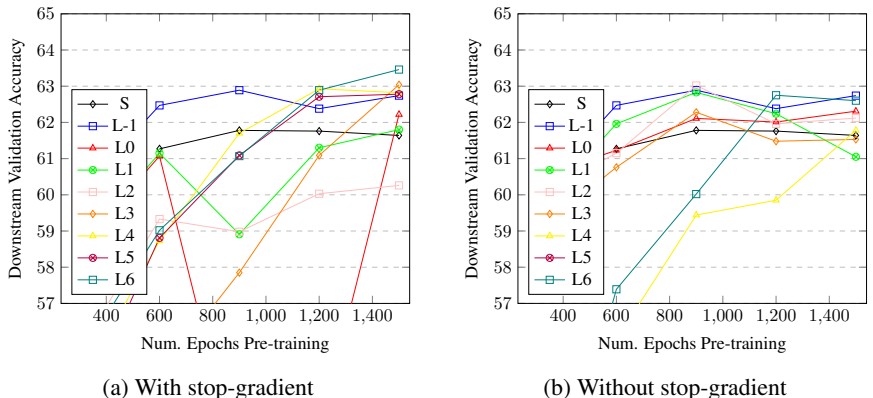

(a) With stop-gradient  (b) Without stop-gradient

Figure 37: PCA CIFAR100: Comparing sampling 50% of batch and subtracting the sixth largest principal component from that sample, with and without stop-gradient.

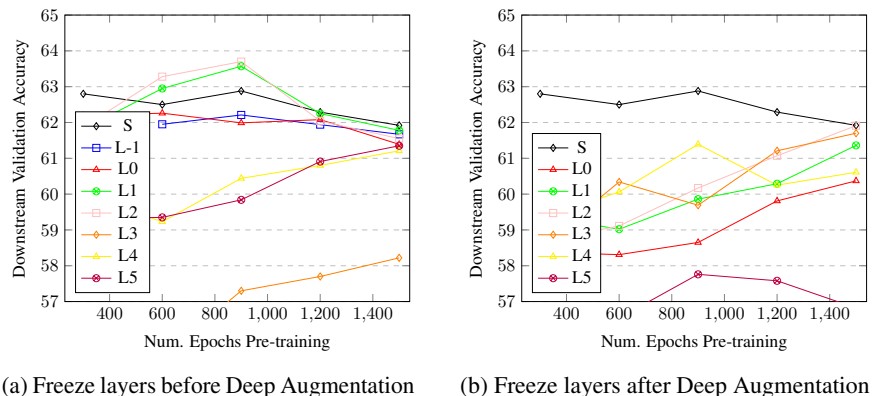

(a) Freeze layers before Deep Augmentation  (b) Freeze layers after Deep Augmentation

Figure 38: CIFAR100. Comparing freezing layers before or after Deep Augmentation with stop-gradient initialized with pre-trained SimCLR model.

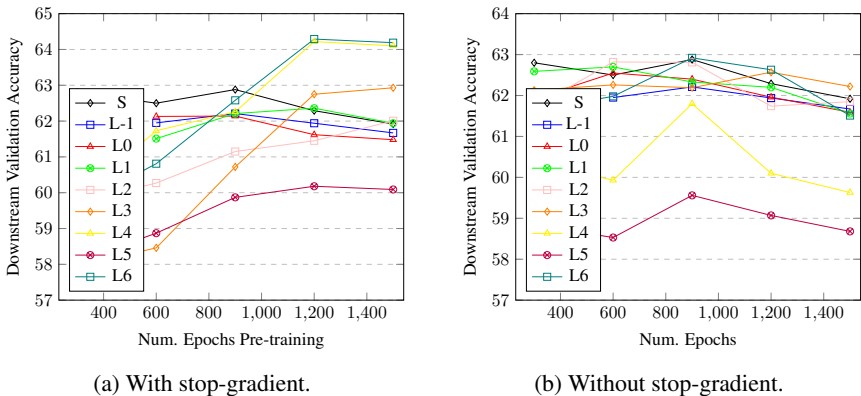

(a) With stop-gradient.

(b) Without stop-gradient.

Figure 39: CIFAR100. 50% sampling, 50% dropout, with and without stop-gradient, and initialized with pre-trained SimCLR model.

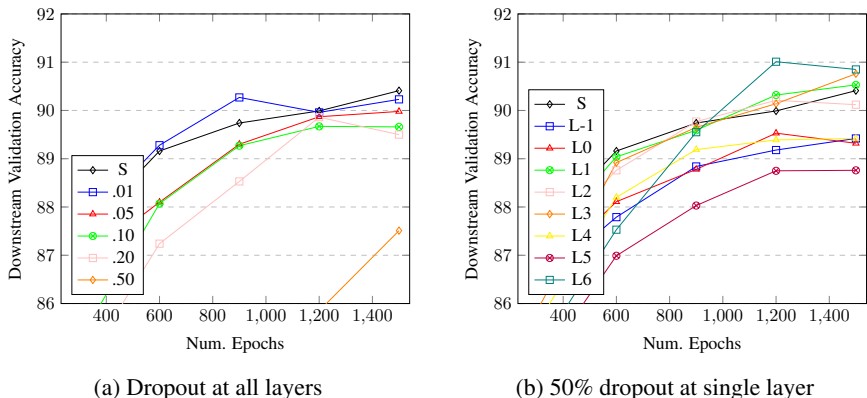

(a) Dropout at all layers

(b) 50% dropout at single layer

Figure 40: CIFAR10. Comparing dropout rates at all layers versus 50% dropout targeted at a specific layer. Note difference in $y$-axis.

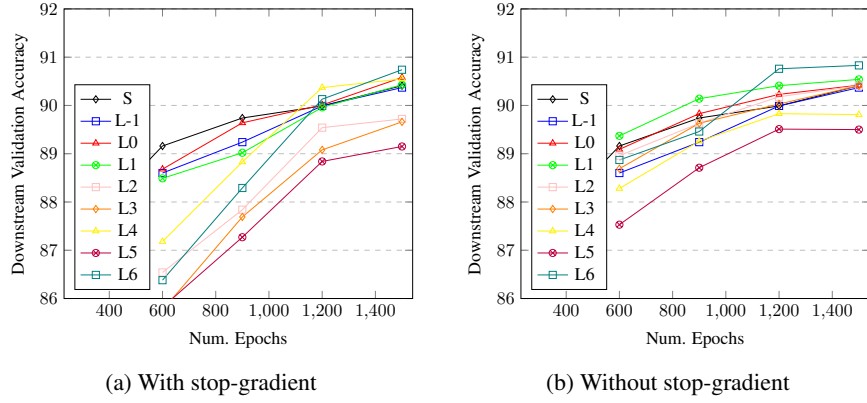

(a) With stop-gradient

(b) Without stop-gradient

Figure 41: CIFAR10. Comparing sampling 50% and applying 50% dropout, with or without stop-gradient.

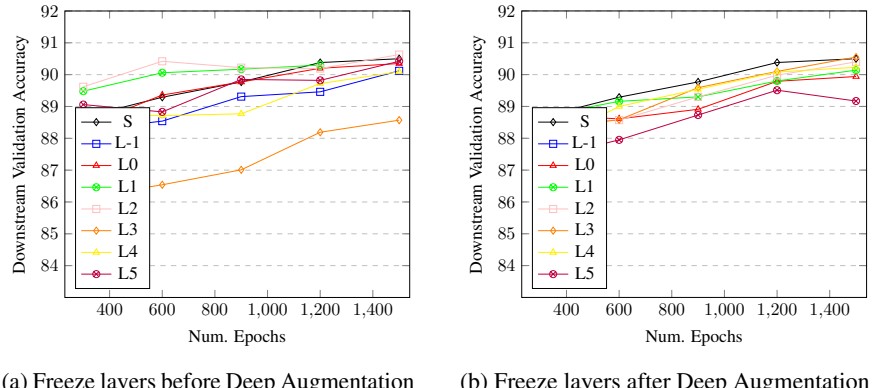

(a) Freeze layers before Deep Augmentation      (b) Freeze layers after Deep Augmentation

Figure 42: CIFAR10. Comparing freezing layers before or after Deep Augmentation with stop-gradient initialized with pre-trained SimCLR model.

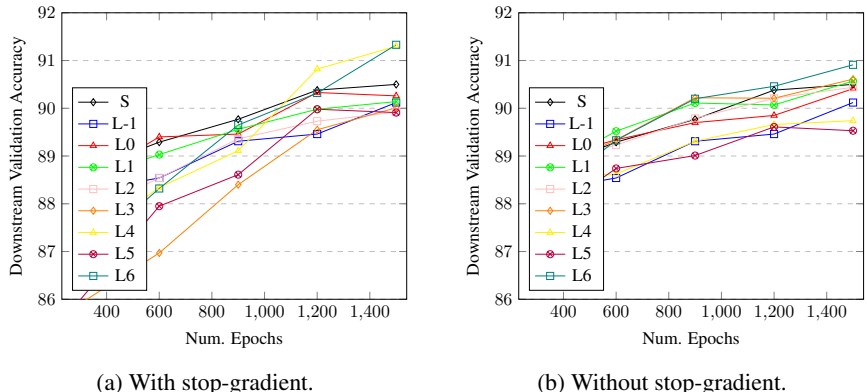

(a) With stop-gradient.      (b) Without stop-gradient.

Figure 43: CIFAR10. 50% sampling, 50% dropout, with and without stop-gradient, and initialized with pre-trained SimCLR model.

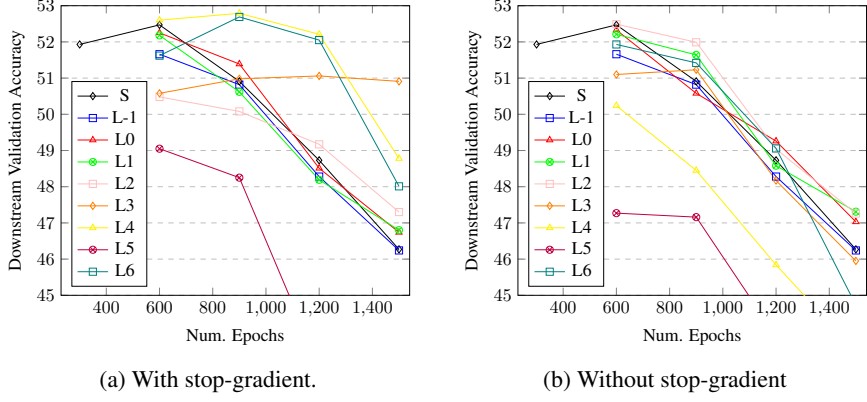

(a) With stop-gradient.      (b) Without stop-gradient

Figure 44: SimCLR and Deep Augmenation with and without stop-gradient pre-trained on CIFAR10 and finetuned on CIFAR100, for different checkpoints during training. Observe the overfitting behavior.

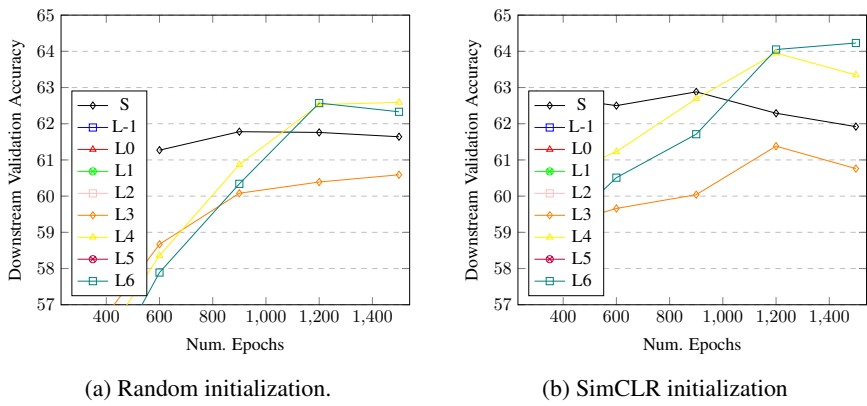

(a) Random initialization.  (b) SimCLR initialization

Figure 45: Deep Augmentation with stop-gradient, only lower-to-higher augmentation pairs.

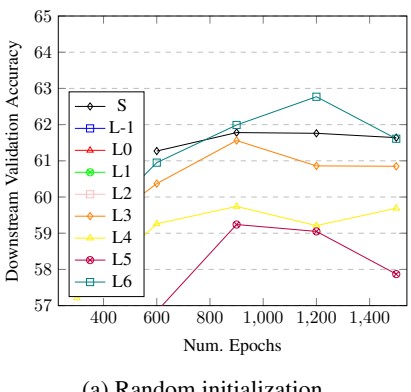

(a) Random initialization.

Figure 46: Deep Augmentation without stop-gradient, only lower-to-higher augmentation pairs.

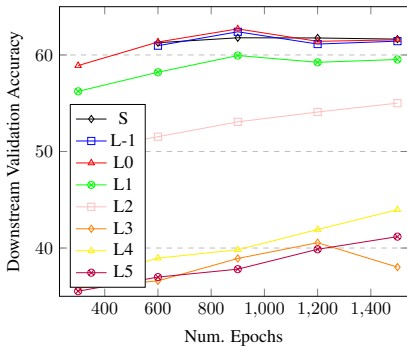

Figure 47: Deep Augmentation with stop-gradient and random initialization, freeze layers before.

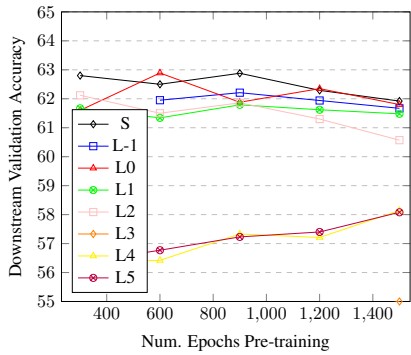

Figure 48: Deep Augmentation with stop-gradient and SimCLR-trained-with-20%-dropout initialization, freeze layers before.

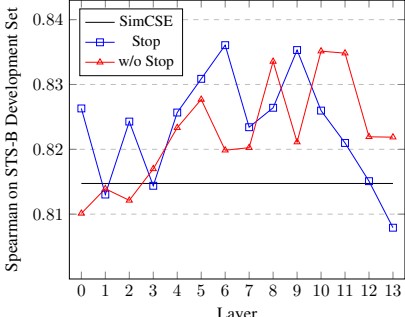

Figure 49: PCA: SimCSE vs. Deep Augmentation to 50% of the batch and subtracting the sixth largest principal component from that sample, with and without stop-gradient. "Stop": stop-gradient. Deep Augmentation outperforms SimCSE..

Table 7: Contrastive Learning on Graphs with GNNs. GCL (Graph Contrastive Learning) versus GCL with Deep Augmentation, dropout, PCA, and with and without stop-gradient.

| Model | COLLAB | IMDB-Multi | NCI1 | PROTEINS |
|---|---|---|---|---|
| GCL | 72.40±0.6 | 53.33±1.1 | 73.97±1.6 | 72.32±1.5 |
| GCL+DeepAug Drop L0 w/ S | 70.93±2.3 | 56.44±3.0 | 73.07±0.7 | 72.92±2.9 |
| GCL+DeepAug Drop L2 w/ S | 70.33±1.5 | 54.00±2.8 | 72.34±0.6 | 71.73±2.3 |
| GCL+DeepAug Drop L4 w/ S | 71.00±1.8 | 52.44±5.1 | 73.32±1.5 | 72.62±1.1 |
| GCL+DeepAug Drop L6 w/ S | **73.80±1.3** | 52.89±4.2 | **75.83±1.0** | 73.21±1.5 |
| GCL+DeepAug Drop L0 w/o S | 71.87±2.7 | **56.89±2.2** | 75.51±1.7 | 73.51±2.6 |
| GCL+DeepAug Drop L2 w/o S | 70.40±2.0 | 52.44±3.9 | 73.32±2.5 | **73.81±2.1** |
| GCL+DeepAug Drop L4 w/o S | 70.93±1.6 | 53.56±3.0 | 75.67±3.3 | 72.32±1.9 |
| GCL+DeepAug Drop L6 w/o S | 70.87±1.1 | 52.00±2.7 | 74.61±1.1 | **73.81±2.3** |
| GCL+DeepAug PCA L0 w/ S | 71.2±1.3 | 54.44±0.8 | 73.4±1.9 | 74.4±2.9 |
| GCL+DeepAug PCA L2 w/ S | 70.53±3.0 | 54.44±2.7 | 73.48±2.4 | 73.21±1.5 |
| GCL+DeepAug PCA L4 w/ S | 70.73±1.4 | 51.11±5.1 | 74.37±0.8 | 72.92±1.1 |
| GCL+DeepAug PCA L6 w/ S | 72.0±0.4 | 50.22±2.2 | 75.59±0.1 | 73.51±0.8 |
| GCL+DeepAug PCA L0 w/o S | 68.13±2.4 | 52.89±3.8 | 74.78±0.6 | 74.4±1.7 |
| GCL+DeepAug PCA L2 w/o S | 70.53±1.3 | 54.0±2.0 | 74.45±0.5 | 72.62±2.3 |
| GCL+DeepAug PCA L4 w/o S | 71.93±0.8 | 54.22±4.9 | 74.21±0.9 | 72.92±2.3 |
| GCL+DeepAug PCA L6 w/o S | 70.87±1.5 | 53.11±0.3 | 75.18±1.0 | 72.02±0.4 |

This poses the following issues for our work: (i) We must be very carefully to compare the Alignment and Uniformity results of sentences with those of images. (ii) Since Deep Augmentation introduces new augmentations, for images, we

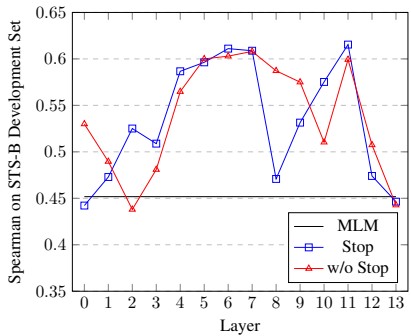

Figure 50: MLM vs. MLM with Deep Augmentation with and without stop-gradient, both without contrastive learning. "Stop": stop-gradient.

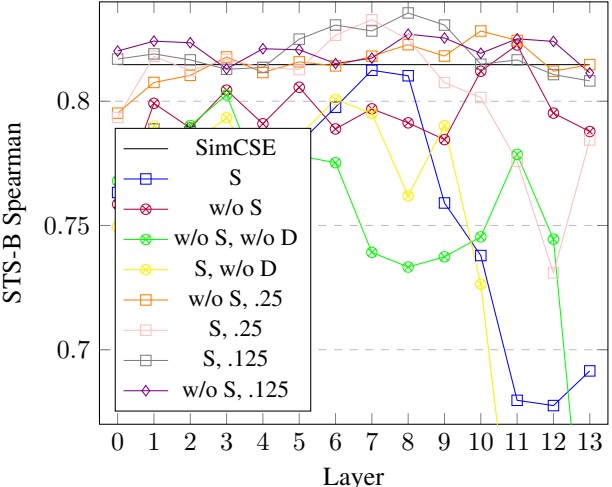

Figure 51: S: short for stop-gradient. D: short for default-dropout, referring to the 10% dropout (including attention-dropout) utilized by BERT and SimCSE. The decimal numbers refer to the Deep Augmentation drop out rate, and is .5 when unspecified.

select only the default data augmentations for the Alignment and Uniformity metrics to enable comparison between Deep Augmentation, baselines, and other settings.

In Table 8, we present the Alignment and Uniformity measures for supervised models on sentence embeddings.

Table 8: Comparison of alignment and uniformity metrics across different models of Supervised Setting on Sentence Embeddings. *: Example of collapse

| Model | Alignment | Uniformity |
|---|---|---|
| Random Init | 0.032 | -0.514 |
| Regular | 0.790 | -2.415 |
| Deep Augmentation L8 w/ stop | 0.734 | -2.318 |
| Deep Augmentation L10 w/o stop | 0.671 | -2.153 |
| Deep Augmentation L1 w/o stop (Fail)* | 0.002 | -0.019 |

# E   CKA Similarity Index Analysis

We include more complete results using CKA similarity index.

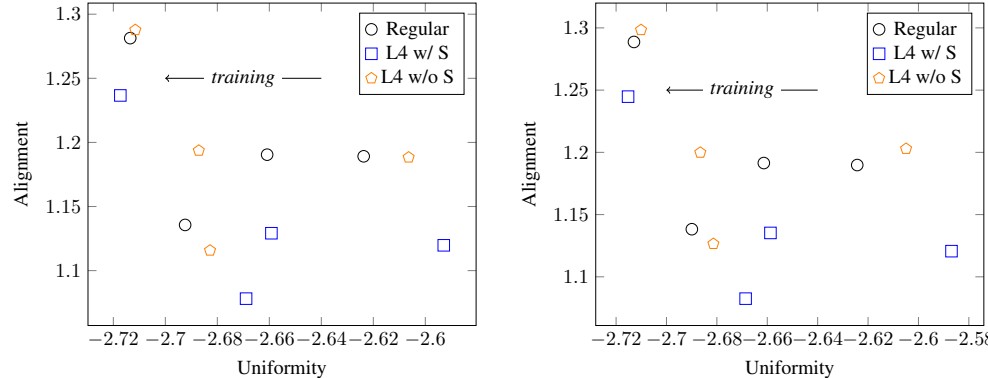

Figure 52: Alignment and Uniformity (lower is better) of Supervised model on SimCLR augmentations on CIFAR. Left: Test data. Right: Training data. Best performance on Alignment and Uniformity does not translate to best performance in supervised setting where the ground truth labels are known. "L" is short for Layer and "S" is short for stop-gradient.

In Figure 53, we include results for several configurations for ResNet18 and CIFAR100. "Layer 4 without Stop" and "Layer 5 with Stop" do not perform well in their downstream performance and share the same increased co-adaptation between layers 4 and 5.

In Figure 55, we include results for several configurations for ResNet18 and CIFAR10. The same trends that were observed on CIFAR100 is also observed on CIFAR10.

Figure 56 displays results for various configurations on sentence embeddings and the STS-B development set. Deep Augmentation achieves optimal performance in the later co-adaptation region, with stop-gradient at its onset and without stop-gradient towards its conclusion.

In Figure 57, we also present the CKA similarity for supervised models. Additionally, Figure 58 shows the CKA similarity for a randomly initialized model. The supervised setting exhibits significantly less co-adaptation between layers, particularly in the later layers. Although Deep Augmentation slightly decreases co-adaptation, this does not correlate with improved performance on the supervised task, suggesting that co-adaptation is less problematic for supervised learning compared to self-supervised learning. We include the "Deep Aug (Fail)" example to illustrate that training collapses, resulting in extremely low accuracy, are associated with very low co-adaptation, indicating that a nuanced level of co-adaptation is necessary to retain information from the data.

It is also worth noting that, since information cannot be created by a deterministic function (i.e., $I(X;Y) \geq I(X;f(Y))$), the reduction in co-adaptation through transformations likely corresponds to a removal of some information from the input data distribution. This suggests that reduced co-adaptation and less overfitting may be linked through the reduction of spurious information in the later layers of the neural network.

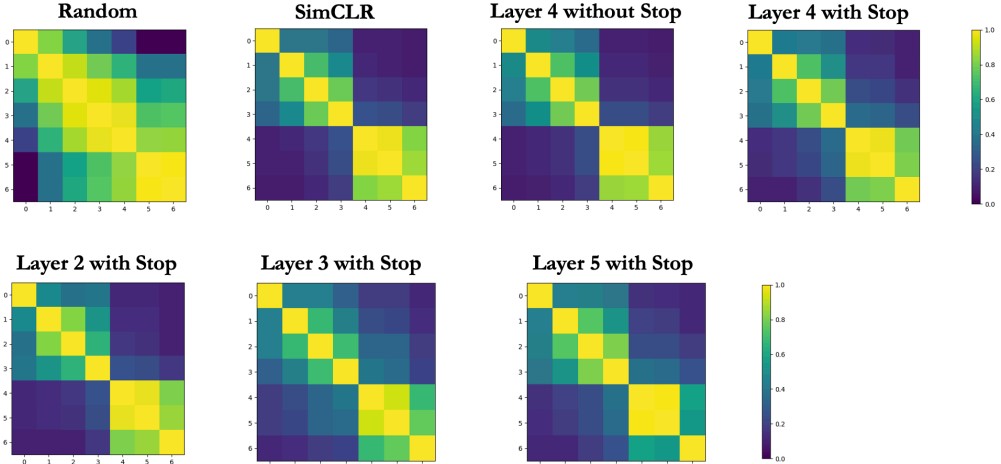

Figure 53: CKA similarity index of ResNet18 for different pre-training methods on CIFAR100.

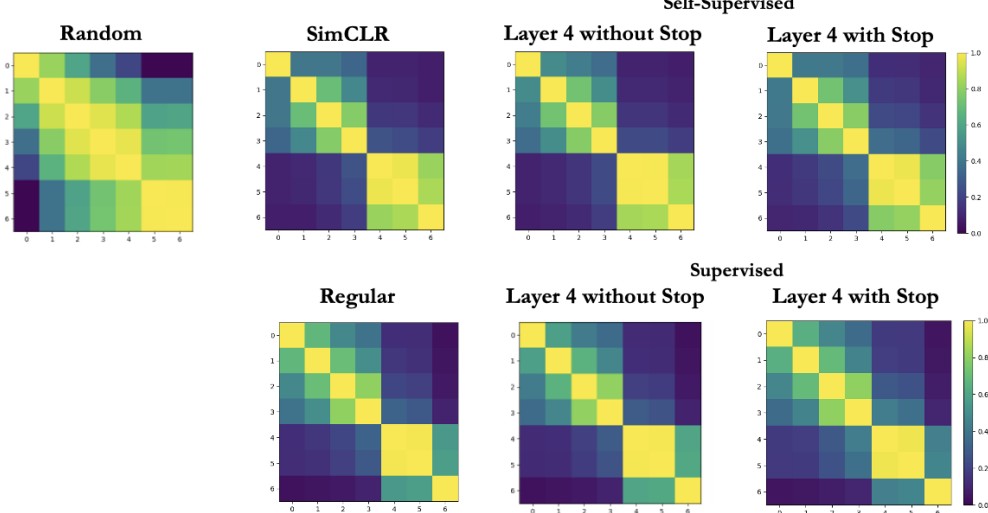

Figure 54: CKA similarity index of ResNet18 for and Deep Augmentation in contrastive learning (self-supervised) versus supervised learning settings on CIFAR100.

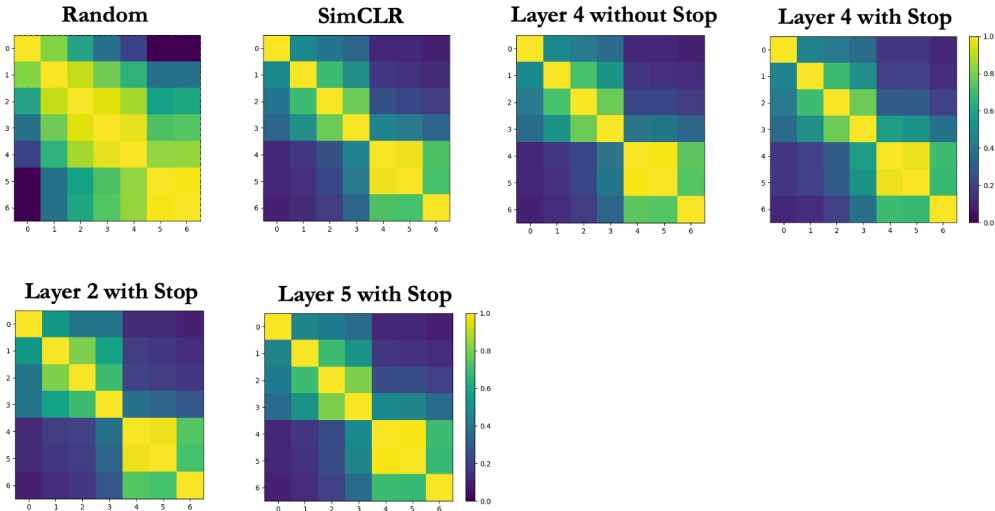

Figure 55: CKA similarity index of ResNet18 for different pre-training methods on CIFAR10.

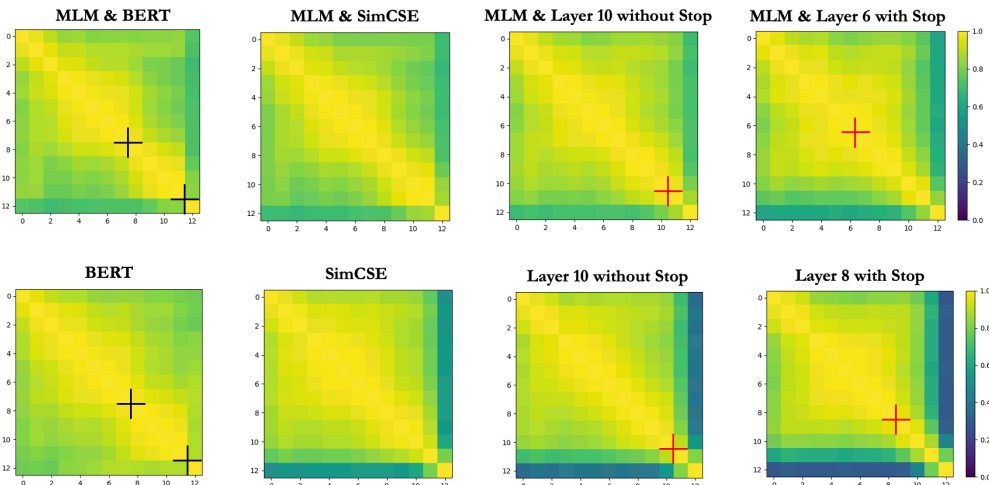

Figure 56: CKA similarity index for different methods trained to produce sentence embeddings. Black crosses indicate the start and end of co-adaptations stretch of layers in BERT, and red crosses indicate where the Deep Augmentation was applied. The layers at which Deep Augmentation performs the best are around the black crosses at the initialization "BERT".

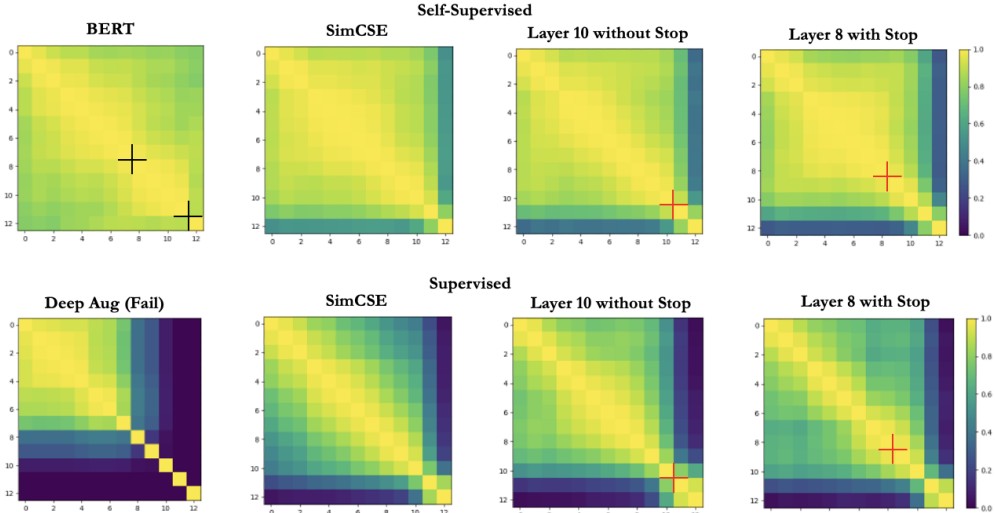

Figure 57: CKA similarity index for different methods trained to produce sentence embeddings. Black crosses indicate the start and end of co-adaptations stretch of layers in BERT, and red crosses indicate where the Deep Augmentation was applied. The layers at which Deep Augmentation performs the best are around the black crosses at the initialization "BERT". Upper row is with contrastive learning (self-supervised) setting and lower row is in supervised setting.

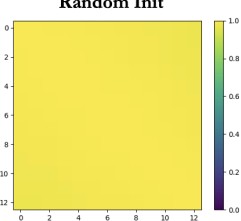

Figure 58: CKA similarity index for a random initialized model.

