# OpenReview forum: "Deep Augmentation: Self-Supervised Learning with Trans- formations in Activation Space"
_TMLR — Rejected by TMLR_

### Review · Reviewer_BtyW · 2024-09-01

**Summary Of Contributions:**

This paper introduces "Deep Augmentation," an alternative augmentation strategy for contrastive learning frameworks that leverages dropout or PCA, with or without the stop-gradient operation. The authors evaluate the performance of contrastive learning frameworks across a diverse set of tasks, with and without Deep Augmentation, and assert that this approach generally enhances downstream performance while reducing computational costs. Additionally, the authors conduct ablation studies on various aspects of the method, including the stop-gradient operation, the placement of dropout/PCA, and the use of pre-trained weights. The study also notes that Deep Augmentation appears to be less effective for supervised learning tasks, possibly due to reduced co-adaptation between neural network layers. Overall, the empirical findings suggest that while Deep Augmentation is a valuable strategy in contrastive learning applications, its effectiveness depends on factors such as neural network architecture and learning paradigms for optimal performance.

**Audience:**

Yes

**Broader Impact Concerns:**

I do not foresee any ethical concerns of this work, and a Broader Impact statement section may not be necessary at this point.

**Claims And Evidence:**

No

**Requested Changes:**

•	The description in Section 3.1 is a bit confusing. To be specific, it is said “… and $z_i^{1}, z_i^{2} \sim \mu$ be a pair of samples”. However, it seems that $z_i^{1}, z_i^{2}$ are just two independent parameters of the augmentation instead of two samples in the dataset. Also, it should be explicitly noted that $A(x_i, z_i)$ represents the augmented sample of $x_i$ after applying the augmentation with parameter $z_i$.

•	The set of indices in the mini-batch should be denoted as $I_b = $ {$1, 2, …, K$} instead of $I_b = 1, 2, …, K$.

•	How should we choose the value of $p$ in the expression of $A_p (x_i, z_i^{j})$?

•	The stop-gradient operation in [1] was mainly used for preventing output collapsing as the authors of [1] posited the problem contains 2 steps of optimization. However, the rationale of using stop-gradient seems to be different in this paper. If so, this should be explicitly explained.

•	Is it mandatory that we should have only one dropout layer to separate the NN into $f_{\theta}^{-1,l}$ and $f_{\theta}^{l+1,L-1}$? What if we have multiple dropout layers?

•	Why do Tables 1, 3, and 4 not have associated uncertainties?

•	What does the dashed line mean in Figure 6(b)?

•	What is the f1mi metric in Figures 8 and 9?

References:

[1]. Chen, X., & He, K. (2021). Exploring simple siamese representation learning. In Proceedings of the IEEE/CVF conference on computer vision and pattern recognition (pp. 15750-15758).

**Strengths And Weaknesses:**

**Strengths**:

•	The proposed Deep Augmentation method is straightforward and easy to understand.

•	Both dropout operation and PCA are model-agnostic and can be applied to many contrastive learning applications if necessary.

•	The authors conduct experiments across multiple domains (NLP, CV, and graph learning) over multiple datasets, which is commendable.

•	The authors study the effect of different components in Deep Augmentation (position of dropout layer, stop-gradient, etc.) through ablation studies, and provide guidance on where to apply the augmentation by conducting similarity analysis between each pair of NN layers.

**Weaknesses**:

•	For all tasks (i.e., NLP, CV, and graph learning) presented in Section 4, the authors only include one standard benchmark contrastive learning framework, which is not quite convincing on demonstrating the effectiveness of the proposed method. More recent contrastive learning benchmarks could be included in the experiments, such as PromCSE [1] for NLP and MCVT [2] for CV.

•	The ablation studies indicate that the performance of Deep Augmentation is highly contingent upon specific hyperparameter configurations. For instance, in CV tasks, the authors recommend combining Deep Augmentation with input data augmentation for optimal results. In graph learning tasks (Figures 8 and 9), it remains unclear whether applying the stop-gradient operation consistently outperforms its absence. Furthermore, the effectiveness of the stop-gradient seems to depend on the position of dropout or PCA operations within the NN architecture. These findings suggest that extensive hyperparameter tuning may be required to achieve optimal performance with Deep Augmentation.

References:

[1]. Jiang, Y., Zhang, L., & Wang, W. (2022). Improved universal sentence embeddings with prompt-based contrastive learning and energy-based learning. arXiv preprint arXiv:2203.06875.

[2]. Mo, S., Sun, Z., & Li, C. (2023). Multi-level contrastive learning for self-supervised vision transformers. In Proceedings of the IEEE/CVF Winter Conference on Applications of Computer Vision (pp. 2778-2787).

---

> ### Author Response · Authors · 2024-10-15
> **Response**
>
> We thank the reviewer for their encouraging and helpful feedback.
>
> **About "More recent contrastive learning benchmarks could be included in the experiments":**
>
> [1] We will reference PromCSE it in our final version of the paper. However, PromCSE deviates from the simple SimCLR setup, and since our focus is not on achieving state-of-the-art performance but on investigating dropout as an augmentation, direct comparison with this work falls outside the scope of our paper. For example, PromCSE uses a different loss function, which would complicate comparisons across datasets and with SimCLR.
>
> [2] MCVT is a significant contribution, and we will mention it in our paper. MCVT applies InfoNCE and ProtoNCE at different stages of Vision Transformer backbones. While it shares similarities with our work in leveraging and exploring intermediate latent spaces, MCVT is specific to transformers and does not focus on targeted dropout or the use of dropout as an augmentation across various data domains, which is the core focus of our work.
>
>
> **Answer to "The ablation studies indicate that the performance of Deep Augmentation is highly contingent upon specific hyperparameter configurations":**
>
> We agree that the number of hyperparameters is a limitation, and our work presents no theoretically foolproof way to select them.
>
> However, we present a practical procedure that consistently works well in our experiments. Dropout combined with stop-gradient consistently improves performance across datasets, including graph, image, and sentence embedding tasks. For graphs, the results with and without stop-gradient are more comparable, but we aimed to show that certain settings work well across all datasets. The trade-offs between using stop-gradient and not using it are not fully understood and require further investigation, although they appear clearer in the self-supervised versus supervised context.
>
> The procedure for selecting the target layer, which we will clarify in the text, is as follows: perform CKA analysis on a network trained without Deep Augmentation, then apply Deep Augmentation with stop-gradient at the first layer in the last co-adaptation region. Without stop-gradient, apply it to the last layer in this region. We recommend using Deep Augmentation with stop-gradient, as it generally performs better across data domains.
>
>
> **Addressing the confusion in Section 3.1:**
>
> We appreciate the reviewer’s observation, and we will revise Section 3.1 for clarity. Specifically, $z_i^1$ and $z_i^2$ represent independent parameters of the augmentation rather than dataset samples, and we will clarify that $A(x_i, z_i)$ is the augmented sample after applying the augmentation with parameter $z_i$.
>
> **Answer to "How should we choose the value of $p$ in the expression of $A_p(x_i, z_i^j)$?"**
>
> In Appendix A.7, we include a plot showing the six largest singular values to illustrate the variance captured. Typically, selecting the largest singular value is effective, although we do not propose a formal procedure for this selection. PCA was included primarily to assess whether dropout is uniquely effective in this context.
>
> **Motivation for stop-gradient:**
>
> Yes, our motivation for using the stop-gradient operation differs somewhat from "Exploring Simple Siamese Representation Learning." As noted in Section 3.2 of our paper: "We apply the stop-gradient operation to halt gradient propagation below the targeted layer during augmentation (Chen & He, 2021). This allows us to investigate the impact of learning invariance to future augmentations, as opposed to those already applied."
>
> **About "Is it mandatory that we should have only one dropout layer to separate the NN":**
>
> It is not mandatory. In this paper, we focus on the effects of single-layer targeted dropout compared to dropout applied at all layers. Investigating multiple dropout layers is indeed an interesting avenue for future research.
>
> **Regarding "Why do Tables 1, 3, and 4 not have associated uncertainties":**
>
> Details and uncertainties for Table 4 are included in the appendix (Table 7). Although Table 3 itself does not display uncertainties, experiments were conducted on both CIFAR10 and CIFAR100, which exhibit consistent trends.
>
> **The dashed line in Figure 6(b)**
>
> The dashed line highlights Layer 5, which is a special, non-trainable layer involving average pooling. Hence, the performance drop at this layer should not be overinterpreted.
>
> **The f1mi metric in Figures 8 and 9**
>
> The f1mi metric represents the micro-averaged F1 score, calculated by aggregating the total false positives, false negatives, and true positives across all classes.

---

### Review · Reviewer_nJin · 2024-09-08

**Summary Of Contributions:**

This paper studies self-supervised learning with contrastive loss. Contrastive-based self-supervised learning uses data augmentation to generate variants of data and learn representations by solving the task of matching variants generated from the same data within a batch or candidate set. Most existing methods use modality-dependent data augmentation to generate variants. Because modality-dependent data augmentations are designed to rely on domain expert knowledge, they are difficult to use in cases where a new modality or domain knowledge is ineffective. The paper proposes a modality-independent data augmentation strategy called Deep Augmentation, which applies dropout or PCA data augmentation at the encoder's intermediate layer. Deep Augmentation does not require domain expert knowledge, so it can be used to apply contrastive learning to arbitrary data sets. The paper examines the effectiveness of Deep Augmentation in three modalities (language, image, and graph); it shows that, in many cases, it achieves performance comparable to baselines that require domain expert knowledge. Furthermore, through ablation and CKA analysis, the paper confirms the effectiveness of the proposed method, especially for contrastive learning.

**Audience:**

Yes

**Broader Impact Concerns:**

Nothing to report.

**Claims And Evidence:**

No

**Requested Changes:**

- Please explain how the proposed method differs from existing modality-independent contrast learning methods. For example, MODAL [a], DACL [b], and the method of Wu et al. [c] seem to address the same problem setting as the paper. Also, in a paradigm other than contrastive learning, Jang et al. [d] propose a method to learn representations by masked-auto-encoding. The novelty and significance of the paper cannot be accurately verified without a discussion of these important related works.
- Please provide experimental or theoretical results comparing the proposed method with the existing methods [a-d] listed above. In particular, DACL experiments should provide a good baseline. Please clarify the superiority of the proposed method over these existing methods.
- Please clarify the design reasons for the proposed method and the rationale for the performance gain. For example, the data augmentation with Gaussian noise introduced in [b] is as simple and effective as the proposed method, so why is it not considered in the proposed method? Also, why is the proposed method superior to SimCSE/SimCLR, which uses domain-dependent data augmentation, in the experiments in Section 4?
- Please explain the data augmentation by PCA in more detail. How does $z^j_i$ apply? Also, why is subtracting principal components an effective data augmentation?
- Please provide a specific algorithm for applying the proposed method.
- If there is a special protocol for selecting the target layer other than the hyperparameter selection method described at the beginning of Section 4, please explain it. If not, then just hyperparameter search is applied, and it should be excluded from the contribution list in Section 1.

[a] Cheung, Tsz-Him, and Dit-Yan Yeung. "Modals: Modality-agnostic automated data augmentation in the latent space." International Conference on Learning Representations. 2020.

[b] Verma, Vikas, et al. "Towards domain-agnostic contrastive learning." International Conference on Machine Learning. PMLR, 2021.

[c] Wu, Huimin, et al. "Randomized Quantization: A Generic Augmentation for Data Agnostic Self-supervised Learning." Proceedings of the IEEE/CVF International Conference on Computer Vision. 2023.

[d] Jang, Huiwon, et al. "Modality-agnostic self-supervised learning with meta-learned masked auto-encoder." Advances in Neural Information Processing Systems 36 (2024).

**Strengths And Weaknesses:**

### **Strengths**
+ The paper presents a simple but effective data augmentation strategy for contrastive-based self-supervised learning.
+ Through the extensive analysis across multiple modalities and datasets, the paper shows the effectiveness of the proposed method.
+ The paper empirically shows that the proposed method can alleviate the co-adaptation problem, which appears in contrastive learning.
### **Weaknesses**
- The paper does not compare the proposed method with the existing contrastive learning method using modality-independent data augmentation, so the novelty and significance of the paper are not sufficiently objectively positioned.
- The proposed method's design is insufficiently valid in terms of theoretical and empirical validity. The paper does not explain why data augmentation with dropout or PCA is superior to domain-dependent data augmentation.
- The paper does not provide a specific contrast learning algorithm based on the proposed method. In particular, Section 1 states that “we propose a procedure for selecting the target layer for Deep Augmentation,” but the reviewer did not find this in the paper.

---

> ### Author Response · Authors · 2024-10-15
> **Response**
>
> We appreciate the reviewer’s thoughtful feedback.
>
> **Clarification of Contributions**
>
> The goal of our study was to answer two key questions: (i) Under what conditions is dropout effective as an augmentation? (ii) Is dropout uniquely effective in these conditions? We address question (i) by evaluating various settings, datasets, target layers, stop-gradient operations, and both self-supervised and supervised learning. For question (ii), we show that PCA augmentation exhibits similar behavior. A comprehensive comparison with other domain-agnostic augmentations was outside the scope of this work.
>
> We do not assert that our method is categorically the superior approach for domain-agnostic augmentation. Our focus is on exploring dropout, a widely used and simple technique in machine learning, as an augmentation strategy.
>
> **Answer to "The paper does not explain why data augmentation with dropout or PCA is superior to domain-dependent data augmentation":**
>
> We do not claim superiority, and indeed these strategies can be used in tandem.
>
> We demonstrate that dropout is a simple and effective alternative/complement to domain-dependent augmentations. Superiority in this context is not a binary concept; rather, our study highlights dropout as a viable augmentation option, complementing existing methods.
>
> **Answer to "The paper does not provide a specific contrast learning algorithm based on the proposed method":**
>
> We will clarify the process of selecting the target layer in the paper. In summary, we perform CKA analysis on the network trained without Deep Augmentation, then apply Deep Augmentation with stop-gradient at the first layer in the final co-adaptation region. For Deep Augmentation without stop-gradient, we apply it at the last layer of this region. We recommend using Deep Augmentation with stop-gradient; empirically, this approach consistently performs well across datasets.
>
> **Related Work Discussion**
>
> We wish to highlight that the mentioned related work is *complementary* to our own, as strategies for data augmentation can be used together.  We are happy to add discussion of these papers to the text, but it would be inaccurate to characterize their contributions as somehow competing with our contribution. **Independently of these related works, we emphasize that our paper already contains extremely extensive experiments that clearly communicate where deep augmentation can be valuable.**
>
> We discuss [a] "Modals: Modality-agnostic automated data augmentation in the latent space" in the paper but will expand that discussion. Modals apply a collection of latent space augmentations, optimized via reinforcement learning. However, none of the augmentations involve dropout. In fact, our augmentations could likely be included in the Modals framework as additional augmentations the algorithm can choose between.
>
> [b] We will add discussion of "Towards Domain-Agnostic Contrastive Learning." It uses MixUp techniques in the input space, whereas we focused on non-MixUp techniques, specifically dropout, in the intermediate latent space.
>
> [c] "Randomized Quantization: A Generic Augmentation for Data Agnostic Self-supervised Learning" is indeed relevant to our discussion. This paper introduces quantization as a novel form of masking along the channel dimension. Both our work and theirs propose augmentation methods--quantization in their case and dropout in ours. Their work emphasizes comparisons to MixUp-style augmentations, while our focus is on ablations and dropout-specific comparisons.
>
> [d] "Modality-Agnostic Self-Supervised Learning with Meta-Learned Masked Auto-Encoder" differs significantly from our approach. [d] focuses on achieving state-of-the-art results using MAE and meta-learning. In contrast, we explore dropout as an augmentation technique, particularly its behavior in supervised versus self-supervised settings. Direct comparison with [d] falls outside the scope of our work.
>
>
> **Answer to "Please clarify the superiority of the proposed method over these existing methods":**
>
> "Superiority" is not the right way to compare these papers.  Augmentation techniques can be used together, so the papers are compementary.  Deep augmentation can be combined with any of the algorithms mentioned by the reviewer to create an even more comprehensive procedure for augmentation.
>
> None of the methods mentioned in the review use dropout as an augmentation technique, highlighting the novelty and interest of our work.
>
> While we conducted experiments with Gaussian noise, we chose not to include them in the paper. We will reintroduce these results.
>
> *On a separate note, please note that the [TMLR acceptance criteria](https://jmlr.org/tmlr/acceptance-criteria.html) do not require state-of-the-art performance or "superiority" as suggested by the reviewer.*

---

> > ### Author Response · Authors · 2024-10-15
> > **Response**
> >
> > **Answer to "Please explain the data augmentation by PCA in more detail":**
> >
> > The term $z_i^j$ introduces randomness into the augmentation, for example, by using different dropout masks for the same data point. In the case of PCA, randomness stems from the stochasticity of the SVD algorithm, and PCA is applied to only 50% of the samples.
> >
> > **Answer to "Please provide a specific algorithm for applying the proposed method":**
> >
> > Could you please clarify what specific details you are looking for? We can provide more details if needed.
> >
> > **Answer to "If there is a special protocol for selecting the target layer other than the hyperparameter selection method described at the beginning of Section 4, please explain it":**
> >
> > As mentioned earlier, we perform CKA analysis on the network trained without Deep Augmentation, then apply Deep Augmentation with stop-gradient at the first layer in the final co-adaptation region. For Deep Augmentation without stop-gradient, we apply it at the last layer of this region.
> >
> > We recommend using Deep Augmentation with stop-gradient; empirically, this approach consistently performs well across datasets. We will include a summary in the paper.

---

### Review · Reviewer_Vusr · 2024-10-01

**Summary Of Contributions:**

This paper tested two main techniques, layer-specific dropout (or PCA) and stop-gradient, as a data augmentation scheme for self-supervised learning. Experimental results from contrastive learning on sentence embeddings, vision, and GNN demonstrated the effectiveness of the proposed method. Further ablation studies pointed out the differences between self-supervised and supervised learning when applying the proposed method. Finally, CKA analysis was employed in order to identify the optimal layer to apply the method.

**Audience:**

Yes

**Claims And Evidence:**

Yes

**Requested Changes:**

Please see weakness above.

**Strengths And Weaknesses:**

Strengths:
1. Good amount of experimental data. Especially in some cases, statistics were provided in addition to just the mean or median, which would be very helpful to convince the readers about the significance of the improvement.
2. Detailed discussion in Ablation Study regarding the potential effect of each factor.


Weaknesses:
1. Target layer selection:
The author emphasizes that the dropout needs to be applied to a specific, optimal layer in order to achieve the optimal results, which is the main differentiation compared to the previous methods. In order to find the "right layer", author suggests to utilize CKA analysis in Section 6.1 to identify the "co-adaptation between layers." However, without providing a more quantitative definition, Fig. 10 may result in very difficult "answers" when interpreted by different researchers. For example, one would argue that whether the author should determine the optimal layer based on SimCSE plot instead of BERT plot, as the augmentation will be applied to SimCSE later. Furthermore, whether one looks at the BERT or SimCSE plot, Layer 3-7 might also seem to be "co-adapted" and could be candidates for the proposed augmentation. But according to Fig. 2a, picking different layers within this range would result in very different amount of improvement. Additionally, one may wonder that if the augmentation truly alleviates the co-adaptation, can the 3rd and the 4th plots in Fig. 10 verify this effect quantitatively? At the end, without a good indicator for the optimal layer, would this method become a trial-and-error kind of optimization technique? (which might be OK, if there's enough empirical evidence to prove the effectiveness...)

2. Generalization:
In Section 5.3 the author suggests that using the proposed dropout together with stop-gradient would be the "good recipe" for all datasets in GNN case. But in Fig 8c and 8d, most of the red bars (w/o stop-gradient) outperform their counterpart blue bars (with stop-gradient) while Fig 8b's optimal case is also red instead of blue. Similar case in Fig. 3, red and blue could both provide comparable optimal results. Readers might wonder if stop-gradient is another trial-and-error knob that could vary case by case?

Overall, the two methods proposed by the author seem to be effective according to the experimental data, but there seems to be lacking good indicators to simplify the optimization process other than brute-force searching.

---

> ### Author Response · Authors · 2024-10-15
> **Response**
>
> We thank the reviewer for their encouraging and helpful feedback.
>
> **Answer to "Target layer selection":** We appreciate your insightful remarks regarding the selection of the target layer. While the method proposed for identifying the optimal layer could be further analyzed in future dedicated work, as shown in Figure 6, the BERT and SimCSE plots exhibit similar co-adaptation patterns, albeit with slightly weaker signals in the SimCSE plot. You are correct that Layers 3–7 also appear to be co-adapted and could serve as candidates for augmentation. Still, our results indicate that applying Deep Augmentation to later layers tends to be more effective.
>
> Regarding your question about whether plots 3 and 4 in Figure 10 quantitatively verify the alleviation of co-adaptation. The plots do not quantitatively verify co-adaptation alleviation across all layers, but they do verify mitigation of co-adaptation in the final layers, which is where we observe the most significant benefits.
>
> **Answer to "Generalization":** Yes, we find that combining dropout with stop-gradient consistently improves performance across datasets, including graphs, images, and sentence embeddings, compared to baseline. While Figures 8c and 8d indicate that the non-stop-gradient condition sometimes performs comparably or better, our goal was to demonstrate that certain configurations work effectively across all datasets.
>
> As you noted, the trade-offs between using and not using stop-gradient are not fully understood and warrant further investigation, though they are somewhat clearer in the self-supervised versus supervised learning context.

---

### Decision · Action_Editor_5JwB · 2024-10-22

**Recommendation:** Reject

**Comment:**

This paper aims to propose/investigate modality/network-independent data augmentation and show that dropout/PCA/stop-grad are reasonable alternatives to classical data augmentation specific to modalities. Overall, adding dropout/PCA/stop-grad improves the performance, whereas the trends are sometimes inconclusive.

All reviewers agree that the proposed method is excellent with its simplicity. The provided analysis through CKA is interesting in its own right.

By contrast, the contributions are unclear at this moment. The introduction emphasizes the proposed Deep Augmentation as the main contribution and claims its benefits by "not require expert-designed and handcrafted augmentations and does not rely on supervised labels, making it versatile and broadly applicable." Nonetheless, the authors claimed that they want to investigate when and how effective dropout is in the authors' response. This may make Reviewer nJin feel that this paper attempts to propose a better method than de facto standards. To avoid such confusion, I would recommend the authors to clearly state their research questions early in the paper.

In addition, we can improve the clarity of evidence; specifically, "co-adaptation" should be defined clearly. Then, we can understand more transparently what expected behaviors are in the analysis of Section 5.

All in all, I do believe that this paper studies an undoubtedly interesting topic. By addressing the above, we can make the manuscript more appealing and solid.

Minor comments:

+ The use of dropout/PCA/stop-grad is more naturally viewed as regularization instead of (implicit) data augmentation. The name "Deep Augmentation" might be misleading in this regard. Although "dropout as data augmentation" was previously claimed, I don't think that this is a common view in the community.
+ If the authors want to put more effort into investigating the underlying success patterns of dropout etc., we could discuss slightly more on the previous investigation in the similar direction. For example, Wu and Gu (2015) discussed how dropout affects differently if inserted in different layers.

Wu, Haibing, and Xiaodong Gu. "Towards dropout training for convolutional neural networks." Neural Networks 71 (2015): 1-10.

**Audience:**

Broadly speaking, the discussed topic is relevant to the broad audience of TMLR. However, as I mentioned in "Claims And Evidence," the main claims of the paper could be made clearer. Otherwise, readers may be unsure whether this paper aligns with their interests.

**Claims And Evidence:**

The manuscript has two issues in this regard.

1. It is unclear whether the authors want to sell the versatility of the proposed method (e.g., layer/network-agnostic nature, modality independence, etc.) or emphasize the contribution to investigate under what conditions dropout/PCA successfully works as an effective regularization. I feel that the submitted version focuses on the former aspect, while the authors' response puts more emphasis on the latter aspect, which makes me confused. Reviewer nJin pointed out the same issue.
2. Supposing that the authors are interested in revealing the mechanism of dropout/PCA, the key phenomenon "co-adaptation" is discussed without its clear definition. This makes it difficult for us to properly assess whether the provided evidence is clear and sufficient. This point is raised by Reviewer Vusr, too.

**Resubmission Of Major Revision:**

The authors may consider submitting a major revision at a later time.